# Cryo-EM structure of photosystem II supercomplex from a green microalga with extreme phototolerance

Rameez Arshad [1,10], Ioannis Skalidis [2,10], David Kopečný [3], Sylva Brabencová [4], Monika Opatíková [1], Petr Ilík [1], Pavel Pospíšil [1], Farzad Hamdi [2,5], Sanja Ćavar Zeljković [6,7], Martina Kopečná [3], Pavel Roudnický [4], Dušan Lazár [1], Eduard Elias [8], Roberta Croce [8], Panagiotis L. Kastritis [2,5,9] & Roman Kouřil [1] ✉

Photosystem II (PSII) is essential for energy conversion during oxygenic photosynthesis in plants and algae. *Chlorella ohadii*, one of the fastest multiplying green algae, thrives under the harsh desert sun but lacks the standard PSII photoprotective mechanisms involving LhcSR/PsbS proteins or protein phosphorylation. Here, we present the cryo-EM structure of the PSII supercomplex from *C. ohadii* at 2.9 Å resolution, which is used to determine whether the exceptional resistance to desert conditions has a structural basis in PSII. The structure reveals a distinct PsbO isoform and additional subunits, PsbR and PsbY, which enhance core complex stability through extensive interactions. Furthermore, the trimeric light-harvesting complexes (LHCII) are bound to the PSII core by specific light-harvesting proteins whose down-regulation in response to high-light conditions implies a reduction in the number of bound LHCII trimers. These structural modifications, together with the high accumulation of specific polyamines in the thylakoid membrane, play a key role in maintaining PSII stability and photoprotection, allowing *C. ohadii* to survive in extreme conditions.

Algae belong to a group of ubiquitous organisms that are present in diverse ecological niches and contribute to about 50 % of global carbon fixation[1]. Algae are often found thriving in extreme habitats, which indicates their remarkable ability to adapt to harsh environmental conditions. In search of stress-resistant algae, an extremophilic microalga, *Chlorella ohadii*, was isolated from the biological sand crust, considered one of the harshest environments in Nature[2]. Due to its ability to survive in challenging environmental conditions, including extreme light intensities (up to 2500 μmol photons m$^{-2}$ s$^{-1}$), large temperature fluctuations, and drought, *C. ohadii* is considered a highly resilient photosynthetic organism[2,3]. This resilience is particularly intriguing because, in terms of photosynthesis, *C. ohadii* lacks classical

[1]Department of Biophysics, Faculty of Science, Palacký University, Šlechtitelů 27, Olomouc, Czech Republic. [2]Department of Integrative Structural Biochemistry, Faculty of Natural Sciences 1 & Biocenter – Biosciences Martin-Luther University, Weinbergweg 22, Halle, Germany. [3]Department of Experimental Biology, Faculty of Science, Palacký University Olomouc, Šlechtitelů 27, Olomouc, Czech Republic. [4]Central European Institute of Technology, Masaryk University, Kamenice 735/5, Brno, Czech Republic. [5]Interdisciplinary Research Center HALOmem, Charles Tanford Protein Center, Martin Luther University Halle-Wittenberg, Kurt-Mothes-Straße 3a, Halle, Germany. [6]Czech Agrifood Research Center, Šlechtitelů 29, Olomouc, Czech Republic. [7]Czech Advanced Technology and Research Institute (CATRIN), Palacký University Olomouc, Šlechtitelů 27, Olomouc, Czech Republic. [8]Department of Physics and Astronomy and LaserLab Amsterdam, Faculty of Science, Vrije Universiteit Amsterdam, De Boelelaan, HV Amsterdam, the Netherlands. [9]Institute of Chemical Biology, National Hallenic Research Foundation, Leof. Vasileos Konstantinou 48, Athina 116 35, Athens, Greece. [10]These authors contributed equally: Rameez Arshad, Ioannis Skalidis. ✉e-mail: roman.kouril@upol.cz

photoprotective mechanisms. It does not possess the LhcSR gene and does not exhibit the typical non-photochemical quenching (NPQ) induction pattern. Although the PsbS gene is present and upregulated under high-light conditions, its role in photoprotection remains unclear, especially when compared to its well-established function in NPQ in land plants and other green algae[4–8]. Hence, studying the structure and function of the photosynthetic apparatus of *C. ohadii* is crucial to understanding the mechanism that allows this alga to grow under such extreme light conditions.

The photochemical reactions of the oxygenic photosynthesis in the thylakoid membrane of cyanobacteria, algae, and land plants are driven by the sequential function of photosystem II (PSII), cytochrome $b_6f$ complex, photosystem I, and ATP synthase[9]. PSII is the catalytic center of the photosynthetic process that drives the water-splitting reaction resulting in the release of molecular oxygen and the generation of protons and electrons, which are subsequently used for the production of ATP and the reduction of $NADP^+$ to NADPH, respectively.

PSII is a membrane-embedded multi-subunit pigment-protein complex formed by the association of the central core complex and the peripheral antenna complexes. The PSII core complex, which consists of a dimer ($C_2$), is evolutionarily conserved in cyanobacteria, algae, and land plants, and contains the subunits and cofactors necessary for the photochemical charge separation, electron transfer, and water splitting reactions[10]. In contrast, the peripheral antenna, which captures and transfers the light energy to the core complex, has become highly diverse during the evolution of photosynthetic organisms[11,12]. In cyanobacteria and red algae, a large membrane-extrinsic protein complex, phycobilisome, serves as PSII antenna[13], whereas the photosynthetic antenna complex of green algae and land plants is formed by membrane-spanning light-harvesting proteins (Lhcbs). In land plants, the PSII antenna is typically composed of 6 Lhcb proteins encoded by the *Lhcb1-6* genes. Lhcb1-3 form homo- or heterotrimeric light-harvesting antenna complexes (LHCII), whereas Lhcb4-6 constitute the monomeric antenna system[14,15]. In green algae, LHCII trimers are formed by various combinations of nine homologous Lhcbm proteins that bind to $C_2$ only through two monomeric antenna proteins, Lhcb4 and Lhcb5, resulting in a partially different organization of LHCII compared to land plants[16–19], except for species from the Pinaceae family[20,21]. Based on the strength of binding of LHCII to $C_2$, we distinguish trimers as S (Strongly), M (Moderately), and L (Loosely, also known as N) bound.

In the past decade, cryo-electron microscopy (cryo-EM) has been utilized to reveal the structural details of PSII supercomplexes from several model organisms of land plants and green algae. The high-resolution PSII structures from spinach and spruce ($C_2S_2$)[22,23], pea[24], and Arabidopsis[25] ($C_2S_2M_2$) revealed the arrangement of subunits and pigments in plant PSII. Further cryo-EM structures of *C. reinhardtii* and other algal species, including *Dunaliella*, diatoms, cryptophytes, and haptophytes, have provided important insights into the core-antenna interactions and energy transfer pathways in algal PSII supercomplexes[16,17,26–35]. Recently, Fadeeva et al[36]. extended these findings for *C. ohadii* by reporting a cryo-EM structure of its PSII supercomplex. Although their study identified additional subunits such as PsbR (also designated as Psb10) and PsbY, their structural roles and binding interactions, along with several other structural features of the PSII supercomplex, were not analyzed in detail or were entirely omitted.

In this work, we present the cryo-EM structure of the $C_2S_2M_2L_2$ PSII supercomplex from *C. ohadii* at 2.9 Å resolution. Our analysis complements and significantly expands the structural data presented in a previous study[36] revealing additional features that may hold the key to the exceptional robustness of this organism in challenging environments, including high-light intensity. In particular, the structure reveals the presence of two additional PSII subunits, PsbR and PsbY, which have not been observed in previously published PSII

structures of other organisms until recently. Our findings, which are consistent with a recent study[36] and further supported by a recently resolved structure of a plant $C_4S_4M_2$ PSII megacomplex[37], suggest that the PsbR and PsbY subunits contribute significantly to the rigidity of the PSII complex and PSII megacomplex. The subunits achieve this by forming numerous interactions within the core complex that play a crucial role in the stabilization of the oxygen-evolving subunits. Further investigation of the PSII structure reveals that the binding of the individual S, M, and L LHCII trimers to $C_2$ is mediated by three specific Lhcb proteins. Based on structural and mass spectrometry (MS) data we conclude that targeted regulation of the expression of these proteins, e.g. under different light conditions, allows optimization of the number of LHCII trimers bound to $C_2$, which plays an important role in PSII photoprotection under high-light conditions. Furthermore, we speculate that the remarkable tolerance of *C. ohadii* to high-light intensity could also be related to the more than tenfold increase in physiologically important molecules known as polyamines. This significant increase in polyamines in the thylakoid membrane may play a key role in the stress response to high-light conditions by stabilizing the photosynthetic apparatus and regulating electron transport. Taken together, these findings point to a distinctive mechanism of photoprotection in *C. ohadii* that relies on the rigid nature of the PSII core, which can efficiently control photosynthetic function and modulate the binding of the peripheral LHCII antenna in response to the light environment. The high-resolution *C. ohadii* PSII structure also allowed us to examine the functional association of various PSII subunits and pigments and determine the pathways of excitation energy transfer.

## Results

### The overall architecture of *C. ohadii* $C_2S_2M_2L_2$ supercomplex

The PSII supercomplexes were isolated and purified from *C. ohadii* grown under normal-light conditions (100 μmol photons m$^{-2}$ s$^{-1}$) by sucrose density gradient (SDG) ultracentrifugation. Single-particle cryo-EM analysis revealed the structure of PSII $C_2S_2M_2L_2$ supercomplex at a resolution of 2.9 Å (Supplementary Figs. 1 and 2). It consists of the dimeric core complex $C_2$, which binds two copies each of S, M, and L-LHCII trimers and two of each monomeric antenna proteins Lhcb4 and Lhcb5 (Fig. 1).

The high resolution of the obtained density maps allowed the modeling of 22 PSII core subunits (in each monomer), 480 light-harvesting pigments, 4 plastoquinone (PQ) molecules, several structural lipids and other cofactors (Supplementary Table 2). The *C. ohadii* PSII structure revealed the presence of two additional subunits, PsbR and PsbY (Figs. 1, 2), which had not been resolved in previously characterized PSII structures from plants and other algae, but were recently observed in the spinach $C_4S_4M_2$ PSII megacomplex structure[37]. The PSII structure contains three membrane-extrinsic lumenal chains of the oxygen-evolving complex (OEC), namely PsbO, PsbQ, and PsbP, which are integral to the functional architecture of the OEC within PSII. The *C. ohadii* PSII structure binds additional chlorophyll (Chl)*a* molecules at previously unreported binding positions in the trimeric LHCII antenna complexes (Supplementary Fig. 3).

The PSII $C_2S_2M_2L_2$ supercomplex from *C. ohadii* has high structural similarity with the known algal type PSII sharing the overall geometry of the core complex and binding positions of light-harvesting antenna[16,17] (Fig. 1). The peripheral light-harvesting antenna is formed by three LHCII heterotrimers (S-LHCII, M-LHCII and L-LHCII) and two monomers (Lhcb4 and Lhcb5) of light-harvesting proteins. Similar to land plants and green algae, S-LHCII and Lhcb5 associate with the PSII core through CP43 and Lhcb4 through CP47. M-LHCII associates with the PSII core through Lhcb4 and L-LHCII binds at the interface of Lhcb4 and CP47. In agreement with a recent study[36], our structural model further confirms the stable association of two additional subunits, PsbR and PsbY, within the core complex. The consistent presence of these subunits in the structure suggests that the

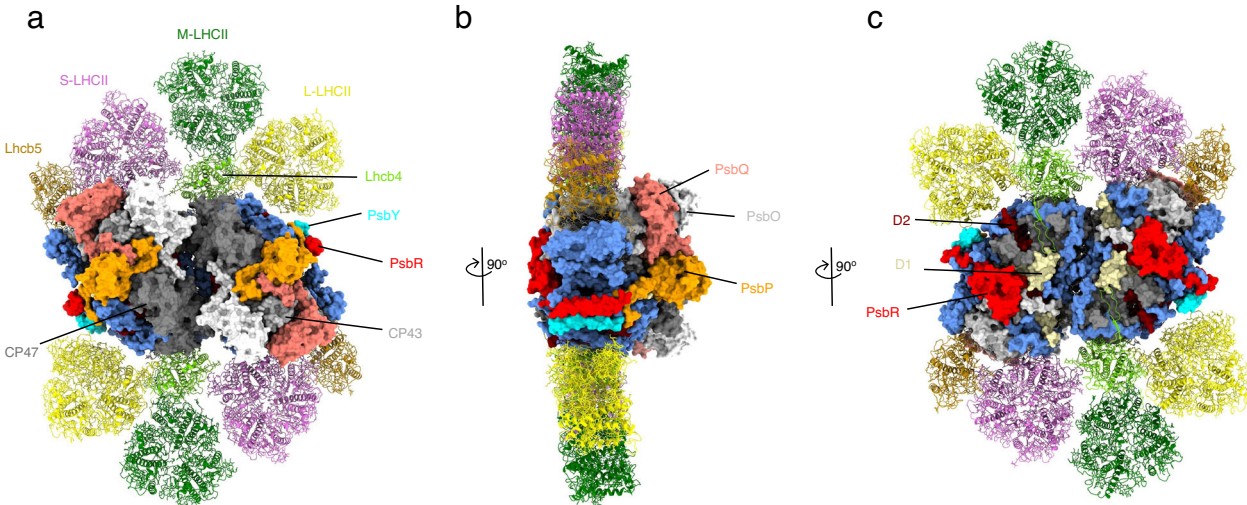

**Fig. 1 | The overall architecture of photosystem II C₂S₂M₂L₂ supercomplex from *C. ohadii*. a** Top view of PSII C₂S₂M₂L₂ supercomplex from the lumenal side. **b** Side view of PSII supercomplex along the membrane plane. **c** Top view of PSII C₂S₂M₂L₂ supercomplex from the stromal side. PSII core complex is shown as a surface model, whereas peripheral antennae are represented as cartoon models. The large core subunits (CP43 and CP47) are shown in light and dark gray in (**a**), whereas the D1 and D2 subunits are shown in light yellow and maroon in (**c**). The subunits of S, M, and L trimers are shown in pink, green, and yellow, the monomeric Lhcb4 and Lhcb5 antennae in lime and gold, respectively. PSII subunits, PsbR and PsbY, are in red and cyan, respectively. Note the extended stromal loop of PsbR in the top view in (**c**). The three subunits of OEC, PsbO/P/Q are visible in the side view in (**b**), indicated in white, orange, and salmon, respectively.

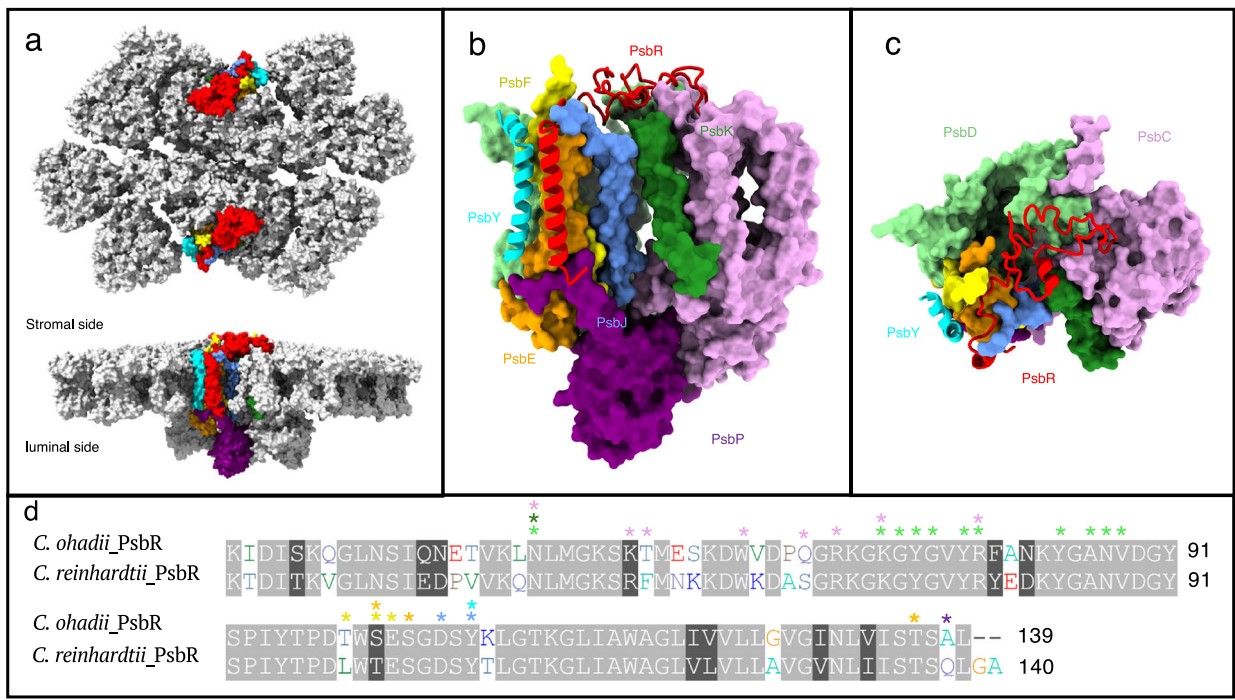

**Fig. 2 | Interactions of the PsbR subunit with the core PsbC, D, E, F, J, K, P, and Y subunits. a** Localization of PsbR subunit in *C. ohadii* PSII supercomplex. **b** Association of PsbR subunit with core subunits PsbC, D, E, F, J, K, Y as observed from membrane plane. **c** Top view of the model from (**b**) showing the interfaces of PsbR with the core subunits on the stromal side of the PSII supercomplex. PSII subunits PsbC, D, E, F, J, K, P, and Y are shown in pink, light green, orange, yellow, light blue, green, magenta, and cyan, respectively. PsbR and PsbY are displayed in cartoon representation in (**b**, **c**), whereas PSII subunits are displayed as surface models in (**a**–**c**). **d** Alignment of amino acid sequences of PsbR protein from *C. ohadii* and *C. reinhardtii* (UniProt ID: A0A2K3DMP5). Conserved amino acids are shown in gray while non-conserved residues are colored. PsbR residues involved in the interaction with core subunits via H bonds are marked with asterisks using the color corresponding to the interacting subunits in (**b**, **c**).

more rigid assembly of PSII is a characteristic feature of *C. ohadii*, which corresponds well with the function of PSII under extreme conditions. This observation is supported by recent findings emphasizing the critical role of structural and functional stability of PSII in high-light tolerance[8].

## Structure of the core complex

In *C. ohadii*, twenty-two membrane-bound subunits of the PSII core were identified, including four large subunits PsbA (D1), PsbB (CP47), PsbC (CP43), PsbD (D2), and low-molecular-weight subunits, namely, PsbE, PsbF, PsbH, PsbI, PsbJ, PsbK, PsbL, PsbM, PsbO, PsbP, PsbQ, PsbT,

Ycf12 (Psb30), PsbW, PsbX, and PsbZ, including two herein localized subunits PsbR and PsbY (see below). Among these PSII core subunits, the structures of PsbO, PsbP, and PsbQ were resolved, and they constitute the membrane-extrinsic components of the oxygen-evolving complex (OEC). Although these subunits are present in all oxygenic photosynthetic organisms, they are frequently absent in the previously published PSII structures[17,22,25,38]. This is primarily due to their relatively weak associations with the core complex, making their structural characterization challenging. In our *C. ohadii* PSII structure, PsbP and PsbQ show structural similarity to their counterparts in other organisms. However, in a recent study of *C. ohadii*[36], the PsbQ subunit was, in our opinion, mislabeled as PsbU, a subunit typical of cyanobacteria that binds to a different site on PSII. This incorrect designation may lead to confusion, as the subunit of interest clearly occupies the canonical PsbQ binding site in its structure. Regarding the third subunit of OEC, we unexpectedly identified and structurally characterized a second isoform of the PsbO protein (denoted by us as PsbO2), which is characterized by a shorter β1–β2 loop (Supplementary Fig. 4). To the best of our knowledge, no such variant of the PsbO protein has been observed in any other PSII structure, including the recent study of *C. ohadii*[36]. It should be noted that land plants such as *Arabidopsis thaliana* also encode two PsbO isoforms, PsbO1 and PsbO2, which are differentially expressed under varying physiological conditions. In Arabidopsis, PsbO2 replaces PsbO1 under stress and has been implicated in the D1 repair cycle during high-light exposure[39,40]. However, the PsbO2 isoform observed in *C. ohadii* represents a distinct structural variant characterized by its shortened β1–β2 loop, suggesting a different adaptation strategy. Because the β1–β2 loop is known to interact with the CP47 protein of the neighboring PSII monomer, it is suggested that the PsbO subunit may contribute to the stabilization of the dimeric organization of PSII[41]. In line with this, the shorter β1-β2 loop of the PsbO2 isoform may reduce the overall flexibility at the interaction interface, leading to an even more rigid conformation that could be crucial for stabilization of the PSII core complex under extreme environmental conditions[39,40].

Moreover, our structure shows that the PQ binding sites of PSII ($Q_A$ and $Q_B$ pockets) are occupied by four PQ molecules. The $Q_A$ site is highly conserved, and a PQ molecule in this site has been found in many PSII structures from different organisms[16,17,23,24]. In contrast, the $Q_B$ site is often empty, or at least the isoprenoid tail of PQ is not resolved in most PSII structures. This absence is likely due to the natural flexibility of the tail and/or weaker interactions in this region. However, our structure demonstrates a stronger PQ binding in both $Q_B$ pockets, which may be attributed to stabilizing interactions of the head part of the PQ molecule within the $Q_B$ pocket. In the recent study of *C. ohadii*[36], the position of PQ in the QB pocket is comparable and residues from D1, D2, and cyt *b559* forming the binding pocket were proposed. However, that study did not provide a detailed description of the nature of the molecular interactions stabilizing PQ. Our analysis revealed that these interactions involve hydrogen bonds (H-bonds) between His215 and Phe265 and the PQ head group. This structural finding is supported by a recent study that demonstrated that Phe265 is critical for PQ binding[42]. Interestingly, in *C. reinhardtii*, due to the slightly different architecture of the $Q_B$ pocket, the latter interaction is mediated with His252 (Supplementary Fig. 5), resulting in a slightly different orientation of the bound PQ molecule. In addition to the head part of the molecule, our study highlights the important role of the D2-Leu43 residue, which interacts with the tail of the PQ molecule and contributes substantially to its stabilization at the $Q_B$ site. Comparative analysis of amino acid sequences in different organisms shows that PSII structures with a completely resolved PQ molecule in the $Q_B$ site consistently contain the D2-Leu43 residue[36,43,44]. In other cases, a replacement of this residue by Phe correlates with the absence or unresolved density of the PQ molecule in the PSII structure[17,38] (Supplementary Fig. 5c). Despite the differences in the residues interacting

with PQ, as shown by the comparison with *C. reinhardtii*, our structure shows that the overall architecture of the $Q_B$ site is relatively conserved. Moreover, our structural findings are in agreement with the mechanistic model previously proposed by Saito et al[45], which highlights the importance of precise PQ positioning and its stabilization for efficient electron and proton transfer at the $Q_B$ site.

## Detailed analysis of PsbR and PsbY subunit integration within the PSII core complex

PsbR, also known as the 10 kDa PSII subunit (Psb10), is present in land plants and green algae and has remained an enigmatic subunit of PSII due to its unknown location and lack of three-dimensional structure. In 2019, Sheng and colleagues[16] suggested a single transmembrane helix near cytochrome *b559* as a potential PsbR subunit in *C. reinhardtii*, although it lacked the N-terminal stromal loop. PsbR has been suggested to be an extrinsic protein functionally associated with PsbP[46–48]. Several studies have suggested considerably diverse roles for PsbR, which include: stabilization of PSII complex, functioning of OEC, role in the PSII repair cycles, binding of stress-related proteins, and formation of stacked PSII[49–52]. Recently, Shan et al. (2024)[37] resolved the PsbR and PsbY subunits in the $C_4S_4M_2$ PSII-LHCII megacomplex from spinach, where they were shown to mediate dimerization of adjacent PSII-LHCII supercomplexes in the thylakoid membrane plane.

Our structure shows that PsbR is a single-helix membrane protein with a long stromal loop containing a short α-helix. These domains extensively interact with several core subunits, including PsbC, PsbD, PsbE, PsbF, PsbJ, PsbK, PsbP, and PsbY (Figs. 1 and 2), and these interactions are mediated by a significant number of H-bonds, salt bridges and also hydrophobic interactions (Supplementary Table 3). On the stromal side of PSII, the position of the N-terminal loop and a short α-helix of PsbR are coordinated by interactions with the PsbD and PsbK subunits through a salt bridge (Lys73$_{PsbR}$-Glu224$_{PsbD}$) and numerous H-bonds. The stromal loop of PsbR further extends into the surface pocket formed by the interfacing loops of PsbC and PsbD, where the interaction occurs via salt bridges (Arg70$_{PsbR}$-Glu456$_{PsbC}$, Lys73$_{PsbR}$-Asp460$_{PsbC}$, Arg79$_{PsbR}$-Asp462$_{PsbC}$) and multiple H-bonds. Subsequently, PsbR winds along the stromal side of PsbD, forming several H-bonds. Close to the peripheral side of PSII, the stromal loop of PsbR interacts with the PsbE and PsbF subunits of the cytochrome *b559* complex, and PsbJ and PsbY, again via a series of H-bonds. In the membrane region, PsbR primarily interacts with PsbE. On the lumenal side, the C-terminal loop of PsbR interacts only with the N-terminus of the PsbP subunit (Supplementary Table 3). These interactions are mostly identical to those observed in the recent PSII structure of *C. ohadii* by Fadeeva et al[36]. However, when compared with those described for spinach PSII-LHCII[37], they appear more extensive, as in spinach[37] PsbR primarily contacts PsbC, PsbD, and PsbJ and forms only a limited number of interactions (Supplementary Fig. 6).

The discovery that the PSII supercomplex from *C. ohadii* contains the complete form of the PsbR protein prompts an intriguing question: Which factors contribute to the stronger binding of this subunit in *C. ohadii* compared to other species? To investigate this, we compared the amino acid sequences of PsbR in *C. ohadii* with that of the model green alga *C. reinhardtii*. This comparison revealed several amino acid substitutions at multiple domains of PsbR in *C. ohadii* (Fig. 2d), specifically at Thr58, Pro67, Gln68, Thr99, and Ala137. These specific substitutions appear crucial, as they are involved in the formation of H-bonds with PsbC, PsbF, and PsbP subunits, potentially enhancing the stability of PsbR binding in *C. ohadii* PSII. In addition, our structural model shows, consistent with the recent PSII structure of *C. ohadii* by Fadeeva et al[36]., that the six core subunits (PsbB, E, F, J, L, X) were resolved with longer N- or C-terminal loops compared to known *C. reinhardtii* PSII structures[16,17] (Supplementary Fig. 7). As the terminal residues of three of these subunits (PsbE, F, J) form H-bonds with PsbR, the PsbE, F, and J loops are less flexible and thus better resolved in the

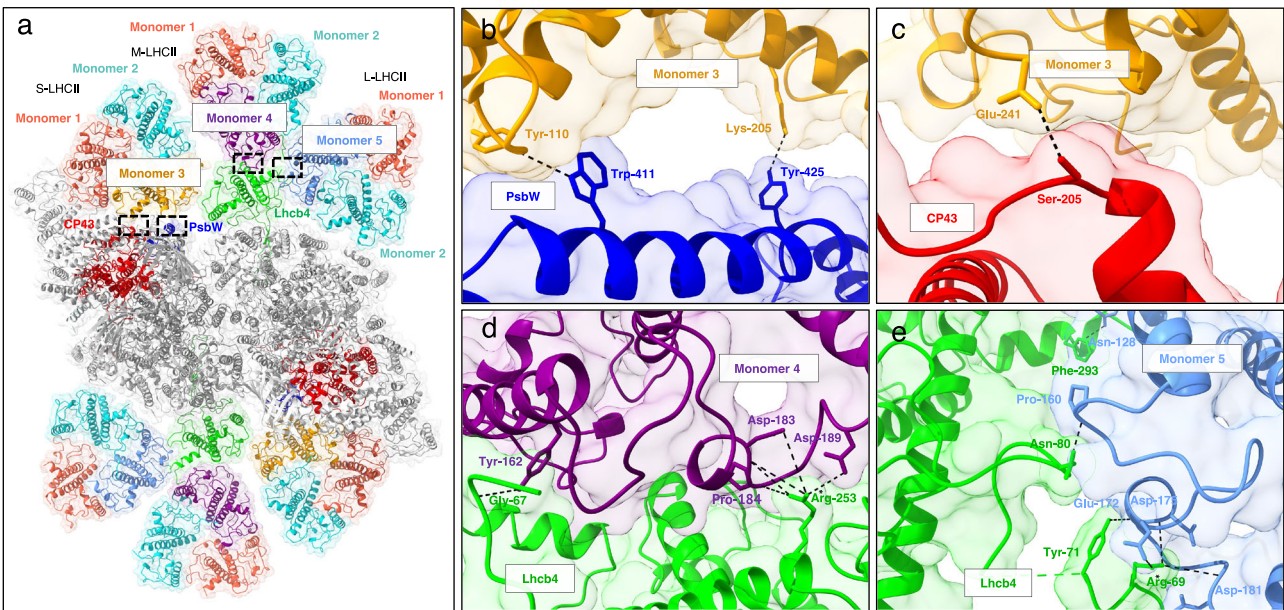

**Fig. 3 | Assembly of peripheral antennae in *C. ohadii* C₂S₂M₂L₂ supercomplex and specific interactions of unique LHCII monomers. a** A view of *C. ohadii* PSII C₂S₂M₂L₂ supercomplex from the lumenal side. The unique monomers of LHCII antenna, monomers 3, 4, and 5, are indicated in orange, purple, and light blue, respectively. The peripheral monomers of LHCII, monomers 1 and 2 are in salmon and cyan, respectively. The core subunits CP43 (PsbC) and PsbW are in red and blue, while the other core subunits are in gray. Lhcb4 protein is shown in green. The black rectangles represent the interface areas of the core and antenna subunits. **b–e** The specific interactions of CP43, PsbW, and Lhcb4 with LHCII antenna monomers 3, 4, and 5. The helices are shown in the cartoon representation and the interacting residues are indicated in the stick model view.

cryo-EM density map. An additional noteworthy finding is the substitution of the N-terminal Tyr7 of PsbF (Leu in *C. reinhardtii*) in *C. ohadii*. The bulky side chain of PsbF-Tyr7 forms two critical H-bonds with residues PsbR-Ser101 and PsbR-Glu102, further reinforcing the interaction between these subunits (Fig.2, Supplementary Table 3). Based on these structural findings, we propose that unique H-bonds formed by specific amino acid substitutions in PsbR and PsbF subunits increase the stability of PsbR binding to the core complex in *C. ohadii*. This contrasts with other green algae and land plant representatives, where PsbR is not as tightly integrated into PSII supercomplexes, presumably due to weaker binding interactions. This observation is consistent with the recent structure of the spinach PSII-LHCII megacomplex, in which PsbR and PsbY were identified only at the dimer interface between two adjacent PSII-LHCII supercomplexes. At this interface, interactions between adjacent supercomplexes appear to locally stabilize the binding of the two subunits. In contrast, on the opposite sides of the PSII-LHCII supercomplexes, which are exposed to the external environment and lack such stabilizing contacts, PsbR and PsbY were not resolved, likely due to their weak binding and subsequent loss during the isolation procedure (Shan et al. 2024)[37].

In the PSII structure of *C. ohadii*, the PsbY subunit, whose location was previously known only in representatives of cyanobacteria[44,53], was also resolved, in agreement with the earlier structure[36]. The *C. ohadii* PSII structure shows that PsbY, like its cyanobacterial counterpart[36], is a single helix membrane protein. However, it is not only associated with the PsbE and PsbF subunits, as observed in cyanobacteria but also forms interactions with the PsbR subunit (Fig. 2, and Supplementary Table 3). In terms of specific interactions, on the lumenal side of PSII, PsbY interacts with PsbE through its N-terminal arginine and tryptophan residues that form a salt bridge (Arg324_PsbY-Asp45_PsbE) and multiple H-bonds. On the stromal side, the C-terminal Gln350 and Asn342 residues of PsbY interact with PsbF and PsbR, respectively, via three H-bonds. A list of interactions of PsbY with other core proteins is provided in Supplementary Table 3. Most of them can also be identified by analysis the recent structure from *C. ohadii*[36]. These interactions suggest a multifaceted role for PsbY in stabilizing the PSII

complex in *C. ohadii*, an aspect not previously observed to such an extent in cyanobacteria. Interestingly, in spinach PSII-LHCII megacomplexes, PsbY also forms similar stabilizing interactions involving Arg and Gln residues with PsbF and PsbE[37] (Supplementary Fig. 6), supporting their conserved structural role across some plant species.

## Assembly of peripheral antennae

The organization of the peripheral light-harvesting antenna of *C. ohadii* resembles the green algal type, characterized by a closely packed assembly of the S-/M-/L-LHCII trimers, which associate with the PSII core complex with the help of Lhcb4 and Lhcb5 (Figs. 1 and 3). In the reported structures of *C. reinhardtii* PSII[16,17], the specific identities of Lhcb proteins forming the M- and L-LHCII trimers remained unclear due to their lower resolution in the cryo-EM maps. In addition to the partial flexibility of the association of M- and L-LHCII trimers to the PSII core, the lower resolution of the trimers was interpreted as a consequence of the heterogeneous composition of these trimers. In contrast, in our density map of PSII from *C. ohadii* the M and L trimers were resolved better, suggesting a specific composition of Lhcb proteins within these trimers. As a result, we were able to unambiguously identify each subunit within the LHCII trimers along with their associated pigment binding sites (see below).

In addition, we identified an additional chlorophyll binding site (Chl *a*615) in LHCII. Specifically, in the S-LHCII, two Chl *a*615 are coordinated by chains N and Y (monomers 1 and 3, respectively; see Fig. 3). Interestingly, the third chain of the S-LHCII (chain G/g, monomer 2) does not have Chl *a*615 in the expected position. In the M- and L-LHCII trimers, due to lower resolution in the peripheral regions, we could model only one Chl *a*615 per trimer (monomers 4 and 5, respectively). Based on our structural model, the Chl *a*615 is coordinated by specific residues at the C-terminus of the Lhcb monomers. The axial ligand appears to be a leucine residue, Leu255, Leu250, and Leu242 in monomers 3, 4, and 5, respectively, while in monomer 1, Chl *a*615 is coordinated by Ser242. Interestingly, we also identified the same coordinating residues in the recent PSII structure of *C. ohadii* by Fadeeva et al[36]., where, due to the better-resolved C-termini in M- and

L-LHCII chains, two Chl $a$615 molecules per each LHCII trimer were resolved. Sequence analysis of the C-termini of the Lhcb proteins that form the S-, M-, and L-LHCII trimers in our structural model shows that specific motifs, such as N**L**A, S**L**A, or **S**AGIGY (coordinating residues in bold), are important for Chl $a$615 binding. In contrast, the Lhcb chain, which consistently lacked Chl a615 in our structure and also in the structure of Fadeeva et al.[36] consistently showed a **FLP**LP motif at the C-terminus that occupies the space of the Chl $a$615 site, effectively preventing its binding. Interestingly, similar motifs, **FTP**SA or **FTP**Q, are found at the C-termini of Lhcb proteins forming S-, M-, and L-LHCII trimers in *C. reinhardtii*[16], where Chl $a$615 was not observed. These results suggest that the specific C-terminal motif of Lhcb proteins plays a crucial role in creating a favorable environment for Chl $a$615 binding and coordination.

Further examination of the LHCII structure in *C. ohadii* revealed that it lacks two out of the three violaxanthin molecules commonly found associated with LHCII trimers of land plants and green algae. This absence could be attributed to the loose binding of these molecules to the antenna monomers, which likely results in their loss during the sample preparation procedure[54]. The absence of violaxanthin was also observed in the digitonin-extracted PSII supercomplex from Arabidopsis, where this phenomenon was correlated with the activity of the detergent[38]. Notably, a recent study of PSII supercomplexes from *C. ohadii* reported an identical deficiency of violaxanthin molecules[36]. Apart from the binding of additional chlorophylls and the absence of violaxanthin molecules, the pigment composition of monomeric and trimeric LHC antenna is similar to the *C. reinhardtii* PSII[17] (Supplementary Table 2).

Based on the cryo-EM density map, we identified five distinct Lhcb proteins (designated as monomers 1-5) (Supplementary Fig. 8) that constitute the LHCII antenna in *C. ohadii*. Their identity was confirmed by the detection of unique peptides in the MS analysis (MS raw data). Sequence comparison of monomers 1–5 with LhcbM proteins from *C. reinhardtii* revealed notable differences in the conserved L18 domain, indicating that these monomers cannot be reliably assigned to the known LhcbM types (I–IV)[55]. In the *C. ohadii* $C_2S_2M_2L_2$ supercomplex, these Lhcb proteins adopt a specific arrangement within the trimeric LHCII antenna. Interestingly, three of these unique Lhcb proteins (designated monomers 3, 4 and 5) occupy the inner positions of the S-, M-, L-LHCII trimers, respectively, whereas the two remaining Lhcb proteins (designated monomers 1 and 2) are located at the peripheral side of each LHCII trimer (Fig. 3). This specific arrangement and specific binding of LHCII heterotrimers in the PSII represent a peculiar aspect of LHCII organization in *C. ohadii* that differs from most land plants and green algae studied to date, where the specific arrangement of LHCII trimers is only partial; for example, in the structure of *C. reinhardtii* PSII, specific Lhcb proteins were resolved only within the S-LHCII[16], probably due to the heterogeneity of Lhcb proteins in -M and -L trimers. Recently, it was shown that the S-LHCII in Norway spruce also has a specific composition consisting only of Lhcb1 protein[22]. In land plants, M-LHCII was found to connect to the PSII core via the Lhcb3 protein. However, analysis of the Lhcb3 mutants revealed that M-LHCII can still bind to the core complex in koLhcb3 PSII[56,57]. This finding implies that the role typically played by Lhcb3 could be substituted by another Lhcb protein, indicating a potential functional redundancy among the Lhcb proteins.

The presence of specific Lhcb proteins at the linker positions of S-/M- and L-LHCII may be important for the photo-regulatory mechanism in *C. ohadii*. In a previous study, Levin et al[6]. demonstrated that photochemically active PSII supercomplexes from *C. ohadii* undergo a significant reduction in LHCII antenna size under high-light conditions[6]. In line with this finding, our *C. ohadii* PSII structure highlights the pivotal role of these specific linker Lhcb proteins (monomers 3, 4 and 5) in determining the size of the LHCII antenna. To further investigate this mechanism of photoprotection, we performed MS analysis on thylakoid membranes isolated from *C. ohadii* cells acclimated to varying conditions: normal-light (NL, 100 µmol photons m$^{-2}$ s$^{-1}$), high-light (HL, 1700 µmol photons m$^{-2}$ s$^{-1}$) and super high-light (SHL, 2500 µmol photons m$^{-2}$ s$^{-1}$). The analysis revealed the down-regulation of two linker Lhcb proteins (monomers 4 and 5) of the M- and L-LHCII trimers under HL and SHL conditions (Supplementary Fig. 9). In addition, the MS data shows that monomer 1 and 2 of each M- and L-LHCII trimers are also downregulated, suggesting that $C_2S_2$ becomes the dominant form of PSII supercomplexes under HL and SHL conditions. This acclimation strategy resembles a response seen in land plants in which $C_2S_2$ represents the physiologically active form under HL conditions[58,59]. On the contrary, the acclimation to HL does not involve the regulation of antenna size in *C. reinhardtii*, however, the antenna size is modulated by the availability of carbon source[60]. On a shorter time scale, regulatory mechanisms such as NPQ and stress-related protein (LhcSR) dissipate the excess energy[61] in *C. reinhardtii*. In contrast, *C. ohadii* employs different photoprotective mechanisms, as neither LhcSR nor NPQ is involved[6,8]. This difference highlights distinct strategies in HL stress responses between these two green algal species.

## Antenna-core and antenna-antenna interactions

In *C. ohadii* PSII, the association of S-LHCII and the PSII core complex is mediated by the subunits CP43 and PsbW with an interface area of 105 Å$^2$ and 130 Å$^2$, respectively. A key lumenal side interaction involves an H-bond between S-LHCII$_{(monomer\ 3)}$-Glu241 and CP43-Ser205 residues (Fig. 3c). The interaction of PsbW with S-LHCII appears to be stronger than with CP43 as it is mediated by two H-bonds involving S-LHCII$_{(monomer\ 3)}$-Tyr110 and Lys205 residues located on the lumenal and stromal sides, respectively (Fig. 3b). Moreover, the monomeric antenna protein, Lhcb4, in *C. ohadii* further supports the association of S-LHCII with an interface area of 91 Å$^2$. The linker proteins of M- and L-LHCII (monomers 4 and 5) interact extensively with the Lhcb4 protein through their helix A and C (AC) loops, forming several H-bonds and salt bridges. On the stromal side, M-LHCII$_{(monomer\ 4)}$-Tyr162, Asp183, Pro184, and Asp189 are in contact with the Gly67 and Arg253 residues of Lhcb4. On the lumenal side, one H-bond between M-LHCII$_{(monomer\ 4)}$-Gly138 and Lhcb4-Asn291 further strengthens the binding of M-LHCII to Lhcb4 (Fig. 3d). The interaction of L-LHCII and Lhcb4 is mainly controlled by four H-bonds and two salt bridges located at the AC loop site of monomer 5 (Fig. 3e). Furthermore, Lhcb5 protein interacts with monomer 1 and monomer 3 of S-LHCII with a surface area of 772 Å$^2$ and 110 Å$^2$, respectively. The association of S-LHCII and Lhcb5 involves several H-bonds and a salt bridge coordinated by Glu172 of S-LHCII$_{(monomer1)}$ and Arg39 of Lhcb5. These interaction patterns are different from what was previously observed in *C. reinhardtii* PSII[16]. The complete list of interactions between light-harvesting proteins and PSII core subunits is provided in the Supplementary Table 4.

The minor light-harvesting antenna proteins, Lhcb4 and Lhcb5, of *C. ohadii* are associated with the core complex and LHCII via different amino acids than in *C. reinhardtii* (Supplementary Fig. 10). The Lhcb4 protein exhibits strong interactions with the core subunits PsbB and PsbH, as well as M- and L-LHCII trimers, primarily via its stromal/lumenal loop regions. Sequence comparison with *C. reinhardtii* reveals that the Lhcb4 protein in *C. ohadii* has unique residues at positions critical for these interactions (Supplementary Fig. 10). At the N-terminus, the algae-specific conserved linker motif RSGGVGYRKY, which is known to be important for connecting M- and L-LHCII, contains substituted residues - Val is replaced by Ala and Lys by Gln. Furthermore, a phosphorylation site (Ser103), present in the long hairpin motif of Lhcb4 in *C. reinhardtii*, is modified in *C. ohadii* by a Gly residue, which therefore, can hamper the previously proposed STT7-dependent phosphorylation and disassociation of Lhcb4[16]. The absence of a Ser103 phosphorylation site is in line with previous

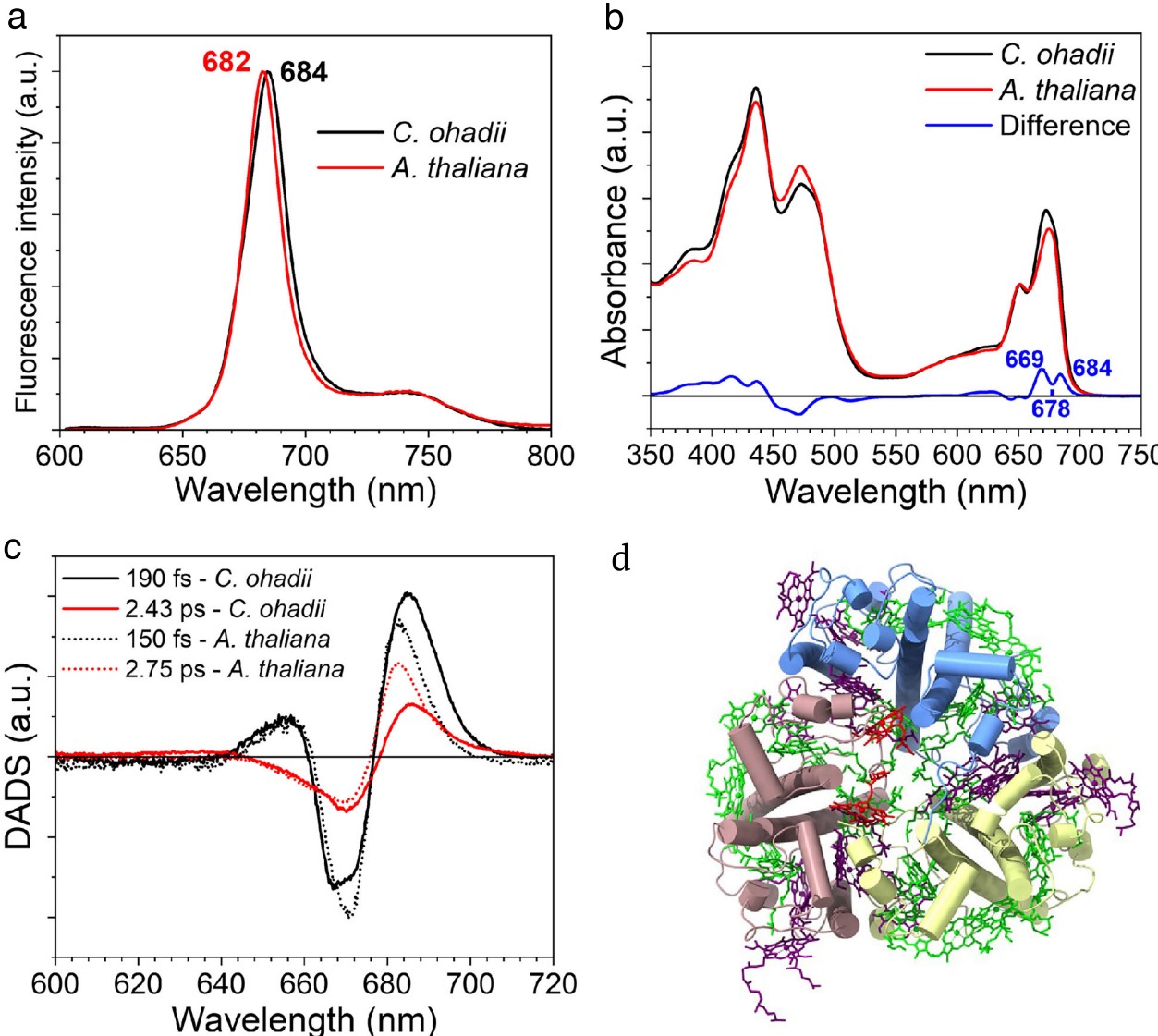

**Fig. 4 | Spectroscopic comparison of *C. ohadii* and *A. thaliana* LHCII trimer.**
**a** Room temperature emission spectra (the excitation was provided at 500 nm).
**b** Absorption spectra. The spectra are normalized to their Qy area (630-800 nm) and scaled to their Chl content, assuming a relative oscillator strength of Chl *a:b* of 1:0.7 and assuming that *A. thaliana* LHCII trimers contain 14 Chls per monomer and *C. ohadii* LHCII trimers 15 Chls. **c** Comparison of the energy transfer DADS of the LHCII trimers excited at 671 nm. The DADS are normalized based on the maximal ΔOD that was measured during the respective TA experiments. **d** Structural model of S-LHCII from *C. ohadii* from the lumenal side of PSII. Positions of Chl *a* and *b* molecules are indicated in green and purple, respectively. Chl *a*615 is shown in red. Source data are provided as a Source Data file.

studies, which report a limited or complete lack of state transition process in *C. ohadii*[6,62]. The Lhcb5 protein also contains unique amino acid sequences at the sites interacting with PsbC, PsbZ, and S-LHCII. Particularly, the N-terminal long loop region, which associates with these subunits, exhibits significant modifications in the interacting residues. In addition, the Lhcb5 protein in *C. ohadii* has undergone a deletion of five residues (compared to Gly33 – Arg37 in *C. reinhardtii*) in the N-terminal long loop region (Supplementary Fig. 10). Hence, we suggest that these distinct amino acid modifications in *C. ohadii* are not just crucial for the binding strength but also the specificity of LHCII (S,M, and L trimer) association with the PSII core complex, indicating a refined adaptation in this species.

### Spectral properties of light-harvesting trimers in *C. ohadii*

We also investigated the spectral properties of the *C. ohadii* LHCII trimers and compared them with a model example of LHCII trimers from *A. thaliana* to assess, in particular, the effect of the additional Chl

*a*615 in *C. ohadii*. The room temperature emission spectrum of *C. ohadii* LHCII peaks at 684 nm, which is red-shifted by 2 nm with respect to the LHCII isolated from *A. thaliana* (Fig. 4a). This is likely the result of a change affecting the lowest energy cluster, that in *A. thaliana* LHCII is formed by the Chls *a* 610-611-612[63–65] (Fig. 4b). The difference absorption spectrum upon normalization to the Chl content of the two complexes (see "Methods") is shown in Fig. 4b. In the Chl *a* Qy region the difference spectrum has positive peaks at 669 and 684 nm and a dip at 678 nm. Considering that the emission data show that the lowest energy cluster is at lower energy in *C. ohadii*, we can assign the dip at 678 nm and the peak at 684 nm to a band-shift feature. The positive 669 nm peak can then be assigned to Chl *a*615 that is present in *C. ohadii* and absent in *A. thaliana*. The 669 nm peak position for this Chl also makes sense from a structural point of view as Chl *a*615 is located on the luminal side, far from the protein interior, and is rather isolated from other Chls *a*, explaining the relatively blue absorption (Fig. 4d, Supplementary Fig. 3).

To determine the effect of the presence of the additional Chl *a*615 on the excitation energy transfer (EET) dynamics of LHCII, we have performed transient absorption (TA) measurements on the LHCII of these two species. We have excited the samples at 671 nm to preferentially excite Chl *a*615 (see Supplementary Fig. 11 for an overlap of the excitation pulse with the absorption spectra of these complexes). Global analysis was performed on the resulting spectro-temporal maps to extract the decay-associated difference spectra (DADS) and evolution-associated difference spectra (EADS). The full set of DADS and EADS is presented in Supplementary Fig. 12. Numerous time-resolved spectroscopy studies have been conducted on LHCII in the past, e.g. ref. [66]–[69]., and the presented data here for both *C. ohadii* and *A. thaliana* is in line with them, showing fast inter-Chl excitation energy equilibration followed by a slow excited state decay. Little attention has however been devoted to the comparison of the ultrafast excitation energy pathways of LHCII across species, with the exception of a few studies[70,71]. A comparison between the energy transfer DADS of the LHCII of both species is shown in Fig. 4c. The time constants of the first components are fairly similar, being 190 fs for *C. ohadii* and 150 fs for *A. thaliana*. In both cases, the DADS show EET from high energy Chls *a* to low energy Chls *a*, which in *C. ohadii* have higher amplitude in the red and peak at 684 nm instead of 682 nm as in *A. thaliana*. This shows that part of this ultra-rapid spectral equilibration process involves the lowest energy Chl *a* cluster as acceptor. The second DADS has a smaller amplitude and describes a much slower EET process, with a time constant of 2.43 ps in *C. ohadii* and 2.75 ps in *A. thaliana*. The main difference is the peak wavelength of the excitation energy acceptors, which is redshifted in *C. ohadii* compared to *A. thaliana*. In the *C. ohadii* DADS there is moreover more negative amplitude in the 668-680 nm region, which we tentatively attribute to EET from Chl *a*615. This also aligns with the structural data that show that this Chl is relatively distant from other Chls *a* and is therefore expected to transfer energy at a relatively long timescale. The last two DADS for both complexes (Supplementary Fig. 12) describe the decay of the Chl excited states, which in a small fraction of the complexes proceeds on the order of 150 ps, but for most of them takes relatively long (>3.5 ns), reflecting the mixture of respectively quenched and unquenched complexes in the ensemble, which is typical for those systems in detergent micelles[72].

## Excitation energy transfer pathways

The precise localization and assignment of chlorophyll molecules in the *C. ohadii* PSII supercomplex allowed us to analyze the main pathways for excitation energy transfer from the individual LHCII trimers, and monomeric antenna proteins Lhcb4 and Lhcb5, to the core complex[73]. The calculated Förster resonance energy transfer (FRET) rates show that the main EET route from LHCII trimers predominantly directs toward CP43 via the S-LHCII trimer (Fig. 5, Supplementary Table 6). Due to very close interactions between the LHCII trimers, the energy absorbed by L-LHCII trimer is transferred mainly to the M-LHCII trimer followed by the transfer to the S-LHCII trimer. As the rates of EET from the L-/M-LHCII and S-LHCII to Lhcb4 and Lhcb5, are two/three times smaller, their contribution is negligible. However, once the excitation energy is localized at Lhcb4 or Lhcb5, it is very quickly directed to CP47 or CP43, respectively (Fig. 5). Comparison of FRET analysis between *C. ohadii* and *C. reinhardtii* shows that both types of green algae utilize a similar strategy for transferring excitation energy from LHCII to the core complex. However, this approach differs from that used by land plants, which transfer excitation energy from individual S- and M-LHCII trimers to the core complex via separate pathways, a difference attributed to their weaker inter-trimer connections. This distinction highlights the different mechanisms of light energy use and transfer in different photosynthetic organisms[73].

## What factors contribute to the high-light resistance of *Chlorella ohadii*?

*C. ohadii* is a green microalga discovered in the desert sand crust that thrives under harsh conditions in its natural habitat[2,3]. The necessity to survive in extreme conditions has led to the evolution of a highly efficient photoprotection mechanism that is still not fully understood. This mechanism is characterized by the absence of the LhcSR and PsbS proteins[6] and NPQ[8], which are typically involved in the photoprotective processes of other green algae. In this study, we show, in agreement with previous reports[6,8,62], that *C. ohadii* growing under HL conditions induces a specific reduction of LHCII antennae size (see Supplementary Fig. 9), an adaptive response that is also not characteristic for green algae. Indeed, the model organism of green algae, *C. reinhardtii*, shows an unchanged Lhcb/PSII core ratio[61] when the cells are grown photoautotrophically. However, even in *C. reinhardtii*, the light-induced response is not uniform and appears to depend on specific environmental conditions, such as high carbon availability, which have been shown to promote a light-induced reduction in antenna size[60]. In contrast, in *C. ohadii*, the modulation of antenna size appears to be strictly light-driven, independent of $CO_2$ availability. This conclusion is supported by the observation that the same light-induced modulation of LHCII antenna size occurred in *C. ohadii* grown under NL and HL conditions in our study, where cells were grown mixotrophically supplemented with additional $CO_2$, and in other studies[6,8,62], where only tris-acetate-phosphate (TAP) medium was used.

In addition to antenna size regulation, our study reveals a significant increase in polyamines, regulatory molecules previously related to metabolic flexibility, and fast growth rates in *C. ohadii*[74]. Polyamines are small organic cations known for their key role in fundamental physiological processes, including growth stimulation, photosynthesis, and enhanced stress resistance, observed in both plants and algae[75,76]. To explore alternative photoregulatory strategies on the level of primary photosynthetic reactions of *C. ohadii*, we examined polyamine levels in thylakoid membranes of *C. ohadii* grown under NL and HL conditions. Analysis by liquid chromatography-mass spectrometry revealed the presence of polyamines in both types of samples analyzed. However, polyamine levels increased more than tenfold in the HL-acclimated sample compared to the NL sample (Supplementary Fig. 13a, Supplementary Table 7). This phenomenon was primarily due to the massive accumulation of the diamines putrescine, cadaverine, and 1,3-diaminopropane in the thylakoid membranes (Supplementary Fig. 13b). This result is consistent with the photoadaptation concept proposed by Kotzabasis and coworkers, which links increased levels of putrescine with the adaptation of photosynthesizing organisms to high-light conditions[77]. This concept includes: (i) a direct action of polyamines on the conformation of LHCIIs[78], (ii) a direct binding of polyamines to the PSII proteins regulating the activity of PSII reaction centers[79], and (iii) alterations in electrical ($\Delta\psi$) and chemical ($\Delta pH$) components of proton motive force[80,81], which regulate photochemical and non-photochemical quenching. Based on the structural and functional analysis we conclude that a rigid PSII core complex, specific antenna adjustment, and efficient regulation of electron transport in thylakoid membrane contribute to the origin of exceptional photodamage resistance of *C. ohadii* to adverse environmental conditions.

## Methods

### Purification of *C. ohadii* PSII supercomplexes

*Chlorella ohadii* cells were cultured in a photobioreactor (Photon Systems Instruments, s.r.o., Drásov, Czech Republic) in a total 3 liters of TAP (Tris-acetate-phosphate) medium bubbled with 2% of $CO_2$ under continuous white light (NL − 100 μmol photons m$^{-2}$ s$^{-1}$, HL − 1700 μmol photons m$^{-2}$ s$^{-1}$ SHL − 2500 μmol photons m$^{-2}$ s$^{-1}$) at 25 °C

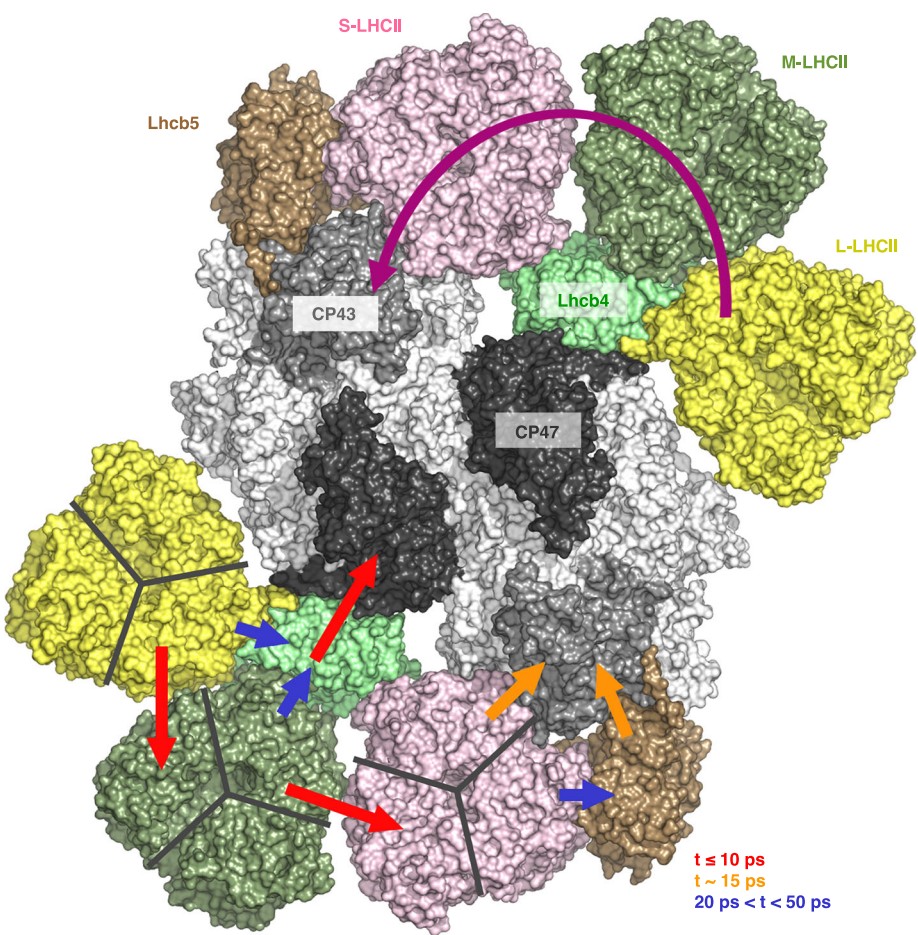

**Fig. 5 | Major energy transfer pathways within the PSII C$_2$S$_2$M$_2$L$_2$ super-complex.** The upper part of the PSII model shows the main overall FRET pathway from the outer LHCII to the PSII core complex (magenta arc arrow), going from the L-LHCII trimer through the M-LHCII to the S-LHCII trimer and to CP43, the inner antenna of the core complex. At the bottom of the PSII model, the arrows of different colors indicate approximate lifetimes of efficient FRET processes from adjacent subunits of LHCII subunits (L-/M-/S-LHCII trimers, Lhcb4 and Lhcb5) towards the core complex (CP43 and CP47). The PSII C$_2$S$_2$M$_2$L$_2$ supercomplex is shown from the lumenal side and the subunits of the oxygen-evolving complex (PsbO-Q) are hidden for clarity. Source data are provided as a Source Data file.

for 4–5 days until an absorbance of 0.7 (measured in a 1 cm cuvette at 720 nm) was reached. *C. ohadii* cells were exposed to HL and SHL for 24 h. The use of elevated levels of CO$_2$ was advantageous for our structural and biochemical analyses of the PSII supercomplex in *C. ohadii* because it allowed us to achieve high cell yields in the smaller culture volume of the photobioreactor (1 L), which was crucial for isolating sufficient amounts of intact supercomplex for cryo-EM and other analyses. The cells were harvested by low-speed centrifugation at 3000 g for 5 mins and resuspended in a buffer containing 15 mM Tris, 15 mM tricine, 15 mM NaCl, 10% sucrose, pH=8.0. The cells were washed and collected by centrifugation at 5000 g for 5 mins. After centrifugation, the cells were resuspended in the above-mentioned buffer and stored at −80 °C. To obtain the thylakoids, the cells were broken by using a microfluidizer device (M110P Homogenizer, Microfluidics) operated at 14 kpsi in repetitive 6 cycles. Unbroken cells and cell debris were separated by centrifugation at 5000 g for 10 mins. Thylakoid membranes were precipitated by high-speed centrifugation (Sigma 3-30KS) at 20,000 g for 2.5 hrs. The membranes in the pellet were resuspended in a minimal volume of cold buffer (15 mM Tris, 15 mM tricine, 15 mM NaCl, 10% sucrose, pH=8.0), which yielded a chlorophyll concentration of 2.15 mg/mL.

Thylakoid membranes were solubilized by 0.65% n-Decyl-α-D-maltopyranoside (α-DDM) at 4 °C for 10 mins with occasional shaking. Unsolubilized material and membrane debris were separated by centrifugation at 12,000 g for 10 mins. The extracted protein complexes were loaded on 10-30% sucrose gradients prepared in MES buffer (25 mM MES, 1 M betaine, pH= 6.5, 0.008% α-DDM) and separated by ultracentrifugation at 284,000 g for 17 hrs at 4 °C in P40ST rotor (Hitachi-Himac). The fractions containing supercomplexes were extracted from the sucrose gradients and stored at −80 °C before further analysis. For cryo-EM analysis, PSII fractions were washed to remove sucrose and concentrated using 50 kDa molecular weight cutoff centrifugal filters (Amicon Ultra-4, Merck Millipore) in MES buffer (25 mM MES, 0.5 M betaine, pH= 6.5, 0.008% α-DDM). The sample washing procedure was repeated 3-4 times and the concentration step was continued to obtain PSII supercomplexes at a final concentration of about 3 mg/mL.

### Purification of *C. ohadii* LHCII trimers

The obtained *C. ohadii* LHCII trimers were concentrated using SDG ultracentrifugation. The sucrose gradients were prepared by thawing a frozen tube that contained a solution of 0.5 M sucrose, 20 mM Hepes pH=7.5, and 0.06% (w/v) α-DDM at 4 °C. The *C. ohadii* LHCII trimers were loaded on top of the SDG and centrifuged at 160,000 g for 18 h at 4 °C. The resulting green band was collected using a syringe. All spectroscopic characterizations were performed on the resulting sample.

### Purification of *A. thaliana* LHCII trimers

*Arabidopsis thaliana* thylakoid membranes were isolated under dim light at 4 °C as in ref. 82. In particular, *Arabidopsis* leaves were shortly blended in a solution (B1) containing 5 mM MgCl$_2$, 20 mM Tricine-KOH

pH=7.8, 5 mM EDTA-Mn, 0.4 M sorbitol and the protease inhibitors 0.2 mM benzamidine, 1 mM e-aminocaproic acid. The solution was centrifuged at 1400 g for 10 min and the pellet was resuspended in a solution (B2) containing 20 mM Tricine-KOH pH=7.8, 0.15 M sorbitol, 5 mM MgCl$_2$, 2.5 mM EDTA-Mn and protease inhibitors as before. This solution was centrifuged at 4000 g for 10 min, the pellet resuspended in 20 mM Hepes-KOH pH=7.5, 15 mM NaCl, 5 mM MgCl$_2$ (solution B3) and centrifuged again at 6000 g for 10 min. The isolated thylakoids were suspended in a solution (B4), containing 20 mM Hepes-KOH pH=7.5, 0.4 M sorbitol, 15 mM NaCl, and 5 mM MgCl$_2$ buffer. A SDG was used to isolate LHCII trimers from the thylakoids as in ref. [83]. In particular, sucrose gradients were created by thawing a frozen tube containing a solution of 0.5 M sucrose, 20 mM Hepes, pH=7.5, and 0.06% (w/v) α-DDM at 4 °C. 200 µg Chl of the sample was diluted to 0.5 mg Chl/ml and then added to an equal volume of solution for solubilization (final concentration: 1% α-DDM in 10 mM Hepes pH=7.5). Unsolubilized samples were discarded after centrifugation at 12,000 g for 10 min, and the supernatant was loaded onto the gradients and centrifuged for 15 h at 4 °C at 160,000 g. The bands were collected with a syringe, and the LHCII trimers were present in the third band.

### Steady-state spectroscopy
Absorption spectra were recorded on a Varian Cary 4000 UV-VIS spectrophotometer. Emission spectra were recorded on a HORIBA JobinYvon-Spex Fluorolog 3.22 spectrofluorimeter at an absorbance of <0.05 cm$^{-1}$ at the Q$_y$ maximum. The absorption spectra were first normalized based on their area in the 630–800 nm region. They were then scaled based on their Chl content, assuming a relative oscillator strength of Chl *a:b* of 1:0.7, and assuming that the *C. ohadii* LHCII trimers contain 15 Chls per monomer, whereas the *A. thaliana* LHCII trimers contain 14.

### Transient absorption spectroscopy and global analysis
Transient absorption measurements were performed using a home-built setup previously described[84]. In short, a Coherent MIRA Ti:Sa mode-locked oscillator seeded a Coherent Rega 9050 regenerative amplifier, which yielded pulses of ~70 fs, centered around 800 nm, at a repetition rate of 40 kHz. Using a beam splitter, these pulses were directed at the probe path (20% intensity) and to the pump path (80% intensity). The pump pulse was converted from 800 nm to the excitation wavelength of 671 nm using a Coherent OPA 9400 optical parametric amplifier. The spectral bandwidth of the excitation was restricted to 10 nm FWHM using interference filters. The probe pulse was directed to a motorized delay stage mounted with a retroreflector, which allowed to measure time delays of up to 3.5 ns. The probe pulse was focused into a YAG crystal to generate supercontinuum white light. In both the pump - and probe path an AA OPTO-ELECTRONIC acousto-optic modulator (AOM) was installed that was used for chopping the pulses on a shot-to-shot basis. The AOMs were triggered by a Stanford Research Systems dg645 digital delay generator that was synced to the regenerative amplifier frequency. The probe pulse was dispersed using a Chromex 250IS spectrograph and its spectrum was recorded using an Entwicklungsbüro EB Stresing CCD camera. The spectra were actively corrected for scatter and dark current. The polarization between the pump - and the probe pulse was set at a magic angle employing a Berek's variable waveplate. The pump-and-probe pulse had FWHM'a of respectively ~150 µm and ~75 µm. The samples were measured in a 1 mm cuvette that was constantly shaken throughout the measurements. Power studies were performed to make sure that annihilation effects were absent in the measurements (see Supplementary Fig. 14). For the measurement presented in the manuscript an excitation energy of 1.5 nJ and 0.6 nJ (at spot size of 150 µm) was used for respectively the *C. ohadii* and *A. thaliana* LHCII trimers.

To extract the main spectro-temporal processes from the transient absorption datasets, global analyses were performed. In the global analysis, the spectro-temporal transient absorption data ψ(λ,t) is fitted to a sum of exponential components in what is called a parallel scheme, yielding the decay-associated difference spectra (DADS), using the following formula:

$$\psi(\lambda, t) = \sum_i \mathrm{DADS}_i(\lambda) \cdot e^{-\frac{t}{\tau_i}} \circledast \mathrm{IRF}(t)$$

In which IRF(t) is the instrument response function. The IRF functions were modeled using a Gaussian function that had an FWHM of 86 fs in the case of the *A. thaliana* experiment and 96 fs in that of *C. ohadii*. The global analyses were performed using the pyglotaran python package[85–87].

### Characterization of PSII subunits and thylakoid membrane proteins by mass spectrometric analysis
Proteins extracted from separated PSII supercomplexes and thylakoid membranes from *C. ohadii* were subjected to filter-aided sample preparation as described elsewhere[88]. The resulting peptides were analysed by liquid chromatography-tandem mass spectrometry (LC–MS/MS) performed using UltiMate 3000 RSLCnano system (Thermo Fisher Scientific) on-line coupled with timsTOF Pro spectrometer (Bruker). See the Supplementary Methods for full details regarding the analyses and data evaluation.

### Estimation of polyamines by LC-MS/MS analysis
Exactly 20 µL of *C. ohadii* membrane sample was mixed with 1 mL of 50% EtOH and sonicated for 10 min. After centrifugation at 14,500 × *g* for 5 min, and 300 µL of supernatant was used for the determination of free polyamines, and another 300 µL for the determination of polyamines conjugated with phenolic acids and other low molecular weight compounds. The remaining pellet was used for the determination of polyamines bound with macromolecules. Derivatization and LC-MS/MS analysis of polyamines were performed according to Ćavar Zeljković et al[89].

### RNA extraction and sequencing of PsbO and Lhcb genes
The total RNA was extracted from *C. ohadii* using the RNeasy Plant Mini Kit (Qiagen). The isolated RNA was treated with Turbo DNAse (Invitrogen) and precipitated by lithium chloride (Invitrogen). The pellet was washed with 70 % ethanol and finally dissolved in DEPC-treated water. The cDNA was synthesized using LunaScript reverse transcriptase (New England Biolabs) with oligo dT primers. The gene coding sequences were amplified by PCR using Q5 High-Fidelity DNA polymerase (New England Biolabs) and Phusion High-Fidelity DNA Polymerase (New England Biolabs) with gene-specific primers (custom-synthesized by Merck) as shown in Supplementary Table 5. Amplified ORFs encoding three PS2 proteins were ligated in the ZERO Blunt vector (Invitrogen) and sequenced.

### Cryo-EM sample preparation and data collection
A sample of the C$_2$S$_2$M$_2$L$_2$ PSII-LHCII complex derived from the sucrose gradient fractionation was initially washed with a washing buffer containing 25 mM MES, 0.5 M betaine, 0.008% α-DDM and with the pH was adjusted to 6.5. In short, to remove sucrose and increase protein concentration, the sample underwent 4 washing rounds using an Amicon filter with a 100 kDa cutoff in a Heraeus Megafuge 40 R (Thermo Scientific), first at 4000 × *g* for 15 min, and then 3 subsequent rounds at 3000 × *g* for 10 min, each time supplementing with fresh washing buffer, until a final sample volume of 65 µl and a concentration of 3 mg·ml$^{-1}$.

The washed and concentrated sample was employed for cryo-EM sample preparation, data collection and analysis. For this purpose,

carbon-coated holey support film type R2/1 on 200 mesh copper grids (Quantifoil) were used. The grids were glow discharged with a PELCO easiGlow, 15 mA, grid negative, at 0.4 mbar and 25 s glowing time. A concentrated sample volume of 3.5 µl was then applied on a grid and vitrified with a Vitrobot Mark IV (Thermo Fisher Scientific) and standard Vitrobot Filter Paper (Grade 595 ash-free filter paper ø55/20 mm). Vitrification chamber conditions were kept stable at 4 °C temperature and 95% humidity. The grids were blotted for 4 s and blot force was set to 0. Following vitrification, the grids were clipped and then loaded into a Thermo Fisher Scientific Glacios 200 keV transmission electron microscope, equipped with a Falcon 4i direct electron detector which was used for data collection. In total, a dataset of 12,419 movies was acquired in linear mode, in EER format, with a total electron exposure dose of 90 e·Å$^{-2}$ using the EPU (Thermo Fisher Scientific) software. Data collection was performed at 150 kx magnification and a pixel size of 0.948 Å px$^{-1}$.

## Cryo-EM data processing and 3D map reconstruction

During data acquisition, the cryoSPARC Live[90] software was employed for on-the-fly analysis. During data streaming, the micrographs were motion corrected and CTF estimated, and a blob picker with a minimum diameter of 200 Å and a maximum diameter of 360 Å was employed. The particles were in parallel extracted with a box size of 820 pix and classified into 100 2D classes. From these classes, 9 were selected automatically with a 6 Å resolution cutoff, containing 72,867 of a total of ~ 2.2 million particles. These particles were then streamed to an ab initio 3D reconstruction, and finally, a homogeneous refinement with C2 symmetry was applied, reaching a final resolution of 3.6 Å (FSC = 0.143). After refinement, the handedness of the map was corrected with ChimeraX[91].

After completion of data collection, the complete set of 12,419 movies was imported in cryoSPARC v4.3.1[92] for single particle analysis. The EER data was split into 30 fractions and then subjected to patch motion correction and patch CTF estimation. Following pre-processing, the final map derived from the on-the-fly analysis was used to create 50 templates that were employed for particle template picking. A dataset of 1,231,582 single-particles was extracted with a box size of 480 pix and subjected to iterative 2D classifications of 200 classes, every time discarding classes containing junk particles, until a final, refined set of 225,372 single particles was obtained. The particles were C2 symmetry expanded, and subjected to a local refinement after taking into consideration their individual CTF parameters, reaching a final resolution of 2.9 Å (FSC = 0.143).

## Model building, refinement, and model analysis

Initial fitting of the subunits in the cryo-EM map was performed by rigid body real-space refinement in Chimera, using the high-resolution crystal structure of PSII from *A. thaliana* (PDB code 7OUI) and Norway spruce (8C29). Local fitting of the subunits in the cryo-EM map was performed using the program Coot. Refinement was performed at 2.95 Å resolution (equal to Fourier Shell Correlation at 0.143) in Phenix (version 1.21.2_5419) using the real-space refinement module and applying geometry, secondary structure, rotamer, and Ramachandran plot restraints. Models of chains G/g, O/o and S/s were obtained by fitting the sequences, which we obtained by cloning respective *C. ohadii* genes (GenBank accessions XHY80384, XHY80383 and XJP35395), into the density map. First, the available sequence for G/g chain (KAI7845556) was excessively long and lacked the correct C-terminus. Second, the available sequence for O/o chain (KAI7845585) lacked the second exon, and thus 31 amino acids were missing. Third, the available sequence for O/o chain (KAI7835897) lacked 39 amino acids from the C-terminus. The validation statistics calculated by MolProbity provided the final score value of 1.85, the overall clash score of 7.9, Ramachandran outliers of 0.0 %, and the CC (mask) /CC (volume) values of 0.82/0.79, respectively (Supplementary

Table 1). Images were prepared with PyMOL. Interactions and sizes of buried surface areas between specific protein subunits and between cofactors were calculated from structural files using the PISA software[93]. FRET analysis was performed according to Sheng et al[16]. and Croce and Amerongen[73]. Representative cryo-EM densities of various ligands bound to the PSII $C_2S_2M_2L_2$ supercomplex are shown in Supplementary Fig. 15.

## Reporting summary

Further information on research design is available in the Nature Portfolio Reporting Summary linked to this article.

## Data availability

The cryo-EM map of *C. ohadii* PSII·LHCII supercomplex generated in this study has been deposited in the Electron Microscopy Data Bank with accession codes EMD-52056. The structure model of $C_2S_2M_2L_2$ supercomplex has been deposited in the PDB under the accession code 9HD7. DNA sequences coding for chains G/g, O/o, and S/s have been deposited to GenBank with accession codes PQ455547, PQ455546, and PQ456901. Mass spectrometry proteomics data have been deposited to the ProteomeXchange Consortium via PRIDE partner repository under dataset identifier PXD059509. Source data are provided with this paper.

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

## Acknowledgements

We thank Toni Kurt Träger for aiding in vitrification experiments and Dr. Christian Tüting for computing infrastructure maintenance. This work was supported by the Johannes Amos Comenius Program—Excellent Research to R.A., M.O., D.K., and R.K. (Registration number CZ.02.01.01/00/22_008/0004624) and by the Grant Agency of the Czech Republic (project no. 23-07744S) to R.A. and R.K. We acknowledge support by the Federal Ministry for Education and Research (BMBF, ZIK program) (grant nos. 03Z22HN23, 03Z22HI2 and 03COV04 to P.L.K.), Horizon Europe ERA Chair 'hot4cryo' project number 101086665 to P.L.K., the European Regional Development Funds for Saxony-Anhalt (grant no. EFRE: ZS/2016/04/78115 to P.L.K.), funding by the Deutsche Forschungsgemeinschaft (DFG) (project numbers: 391498659 (RTG 2467); and (SFB 1664, TP D1, A4, C4) to P.L.K.), and the Martin-Luther University of Halle-Wittenberg. S.B. was supported by the project National Institute of Virology and Bacteriology (Program EXCELES, ID Project No. LX22NPO5103) - Funded by the European Union - Next Generation EU. The work by S.Ć.Z was funded by project MZE-RO0425 from the Ministry of Agriculture, Czech Republic. The work at VU Amsterdam was supported by the Dutch organization for scientific research via a TOP grant to R.C. We acknowledge CEITEC Proteomics Core Facility of CIISB, Instruct-CZ Center, supported by MEYS CR (LM2023042, e-INFRA CZ (ID:90254)).

## Author contributions

R.A., M.O., P.I., and R.K. designed the study. R.A., I.S., S.B., M.O., sample preparation for cryo electron microscopy. F.H. collected the high-resolution cryo-EM data. I.S., P.K. image analysis of cryo-EM data. D.K., R.A., R.K. model building and analysis. P.R. mass spectrometry analysis, E.E., R.C. spectroscopy analysis, S.Ć.Z. polyamine analysis, P.P. pigment analysis, M.K. RNA extraction and gene sequencing. R.A., D.K., I.S., P.I., D.L., R.K. data interpretation. R.A. and R.K. wrote the main body of the manuscript and all authors revised and approved it.

## Competing interests

The authors declare no competing interests.
