## [Peer review File · Nature Communications]

Cryo-EM structure of photosystem II supercomplex from a green microalga with extreme phototolerance

Corresponding Author: Dr Roman Kouril

Version 0:

Reviewer comments:

Reviewer #1

(Remarks to the Author)

Dear Authors,

This is interesting comprehensive study of PSII-PHCII supercomplex from green microalga *Chlorella ohadii* known for its extreme phototolerance. The manuscript primarily focuses on structural determination of the supercomplex with application of cry-EM but also presents its spectroscopic properties. It is original, significant work noteworthy of publication in Nat Comm.

However, in my opinion the current form of the manuscript requires some important reevaluations before it will be ready for positive recommendation for publication.

In the areas in which I feel competent I found some flaws that I kindly ask authors to address.

These are detailed below. The pagination and line numbering are according to what I received in reviewer's pdf.

Page 3, lines 86-91: "This structure reveals several novel features that may hold the key to the exceptional robustness of this organism in challenging environments, including high light intensity. In particular, the structure reveals the presence of two additional PSII subunits, PsbR and PsbY, which have not been observed in previously published PSII structures of other organisms. Our findings, which are consistent with a recent study²⁴, suggest that the PsbR and PsbY subunits contribute significantly to the rigidity of the PSII complex."

Page 3, lines 116-118: "The *C. ohadii* PSII structure showed the presence of two additional subunits PsbR and PsbY (Figs. 1,2), which have not been identified in the previously characterized PSII structures from plants and other algae."

Very recently PsbR and PsbY subunits were resolved in cryo-EM structure of C4S4M2 PSII-LHCII and are published (Shan, J.; Niedzwiedzki, D. M.; Tomar, R. S.; Liu, Z.; Liu, H., Architecture and functional regulation of a plant PSII-LHCII megacomplex. *Sci Adv* 2024, 10, eadq9967, DOI: 10.1126/sciadv.adq9967).

Abovementioned parts of the manuscript should be adequately amended. Moreover, since the referred above paper extensively elaborates on roles of PsbR and PsbY subunits, I suggest a careful reevaluation of the manuscript section titled "Detailed analysis of PsbR and PsbY subunit integration within the PSII core complex" on Pages 5-6, Lines 182-231.

Steady-state spectroscopy

Page 15, lines 501-503: "Absorption spectra were recorded on a Varian Cary 4000 UV-VIS spectrophotometer. Emission spectra were recorded on a HORIBA JobinYvon-Spex Fluorolog 3.22 spectrofluorimeter at an optical density of <0.05 cm⁻¹ at the Qy maximum".

Considering what was written in this section of the manuscript, "optical density" value at Qy band of Chl a was directly used to calculate sample concentration (Chls content). Therefore, it assumes it is scattering free sample. Thus, more appropriate will be to use term Absorbance instead of Optical Density. The latter one incorporates also sample reflectance and scattering which as it sounds are negligible here.

Page 11: Figure 4 a and b: Absorption and Emission are physical processes and such names should be avoided on Y-axis to describe strength of the emission or absorption signal. I suggest changing them to "Fluorescence intensity" and "Absorbance", respectively.

Transient absorption spectroscopy

Page 15 line 524: "Power studies were performed to make sure that annihilation effects were absent in the measurements."

The sentence mentioned that these studies were done but results are nowhere to be found to validate them. Provide (in SI) a graph with exemplary TA trace probed at some specific wavelength (like 680 nm or so) for few different excitation powers/intensities. Normalize them at maximum to show that dynamics are not affected and overlap.

Moreover, there is no information what excitation power (or energy, or fluence, and excitation beam size on the sample) was used to obtain results presented in this work.

Transient absorption of LHCII trimers

Page 10-11, lines 366-381, Figure 4C:

I do understand motivation for comparison of TA of LHCII from both species. However, it should be also discussed in perspective to what is known on this subject from past studies. This is not a novel subject. There is a vast number of papers on it and not even one was referred here.

Why last two DADS were taken out of the analysis shown in Figure 4c? It is convenient but readers would like to know what they represent. Based on TA data description, discussion and methodology one would assume that authors present annihilation free results. I am not so convinced. The 150-200 fs and 2-3 ps DADS are indeed associated with EET process within monomer (positive-negative shape) however 130-170 ps and ~4 ns DADS have the same (only negative) profiles of very different amplitudes. It mean these are mutually independent but share the common spectral shape. Since 4 ns DADS is associated with intrinsic decay of Chl a excited singlet state what 130-170 ps DADS does represent? In my opinion it is single-singlet annihilation on monomer-monomer LHCII interface within the trimer.

I think all these will play out and will be much better seen if instead of the most basic parallel excitation decay model a more intuitive sequential scheme is used for TA fitting. I am sure it could be easily implemented in pygotaran. Both models produce macroscopic rates (so lifetimes do not change) but the sequence gives spectral evolution of the bleaching of Qy band of Chls. Such presentation will be substantially more digestible for readers who may clearly see resemblance of Qy band absorption and its bleaching and how it changes over delay time from moment of sample excitation. This is not so easy to decipher from meaningless shapes of DADS profiles. For visualization I encourage authors to look into this paper (Gradinaru et al. Selective Interaction between Xanthophylls and Chlorophylls in LHCII Probed by Femtosecond Transient Absorption Spectroscopy, <https://doi.org/10.1021/jp026278q>) and particularly focus on Figure 2B and accompanied discussion on singlet-singlet annihilation.

Is 671 nm excitation beam profile sufficiently narrow to excite only Chls a in the LHCII trimer or it will also partially overlap with Chl b absorption? Could authors add one more figure in the SI that shows both, Qy band absorption of LHCII and laser pulse spectrum, so there will be clarity about it.

Reviewer #2

(Remarks to the Author)

The study focuses on understanding the remarkable phototolerance of the green microalga *Chlorella ohadii*, which thrives in extreme desert conditions despite lacking conventional photoprotective mechanisms such as NPQ and stress-related proteins (LhcSR, PsbS). The research goal was to elucidate whether the exceptional resistance of this alga to high light conditions has a structural basis within the Photosystem II supercomplex. Using single-particle cryo-EM, the structure of the PSII supercomplex (C2S2M2L2) was determined to 2.9 Å resolution.

Key structural findings apart from the discovery of a unique PsbO isoform, are not entirely novel, as a very similar structure of the photosystem II C2S2M2L2 supercomplex from *C. ohadii* was already published by Fadeeva et al. (2023, PDB 8BD3, reference 24 in the current article). The roles of subunits PsbR (Psb10) and PsbY as photoprotectors, as well as their stabilizing interactions with cytochrome b559, were previously proposed. The additional chlorophyll-binding site (Chl a615) in LHCII was also present in the earlier 8BD3 structure. It is concerning that readers might sense the authors are attempting to obscure the previous publication rather than clearly stating the differences between the two structures. The authors need to explicitly highlight how their results represent clear advancements or novel insights beyond those previously reported, specifically addressing comparisons with Fadeeva et al.

Additionally, specific Lhcb proteins within the antenna complexes were found to modulate the binding of light-harvesting antenna trimers in response to light conditions, supported by mass spectrometry data showing downregulation under high-light conditions. The study also found a significant increase in polyamines in high-light acclimated samples, suggesting a photoprotective role. Collectively, the results suggest a distinct photoprotection strategy in *C. ohadii*, relying on a rigid PSII core, specialized antenna composition, and polyamine-regulated electron transport and stability of the thylakoid membrane. Although the structure was presented as the primary result of the manuscript, the manuscript's real strength lies in additional analyses such as mass spectrometry, spectroscopy, LC-MS/MS, RNA-seq, and FRET, particularly given that Fadeeva et al.'s publication relied solely on structural data. Hence, the focus of the manuscript might need to be revised by the authors. Line 74- The sentence discussing LHCII trimers should cite algae PSII structures and not only plant structures.

Line 82 - The sentence referring to cryo-EM structures of *C. reinhardtii* should mention all known PSII structures from algae, including *Dunaliella* and *Chlorella*.

Line 86 -90. The authors claim novel features that have not been observed previously, yet this overlaps significantly with Fadeeva et al. (2023).

Line 302 -suggested editing: In contrast, *C. ohadii* employs different photoprotective mechanisms, as neither LhcSR nor NPQ is involved

Line 422 - see Extended Data Fig. 8 and citations of papers by Isreal groups"—this should explicitly refer to specific citations.

Line 435 – suggested editing: However, polyamine levels increased more than tenfold in the HL-acclimated sample compared to the NL sample

References 24 and 38 are duplicates and need correction.

What was the reason for comparing the spectroscopic data to *Arabidopsis* rather than to a more closely related alga, such as *Chlamydomonas*? Likewise, what was the reason for comparing PsbO to diatom and red algae? I suggest adding sequences from the green lineage.

The discussion lacks a clear connection between specific structural features and photoprotection.

Discrepancies exist between chlorophyll numbers mentioned in the text and those found in the structural model—for example, chlorophyll 615 of the PsbN subunit is absent from the provided model cif file.

Reviewer #3

(Remarks to the Author)

In the manuscript presented by Arshad, R. et al., the cryo-EM structure of a PSII-LHCII supercomplex from a high-light-tolerant green alga named *Chlorella ohadii* was solved and provided several interesting features including two small intrinsic subunits named PsbR and PsbY as well as a newly identified chlorophyll molecule found in the luminal cavity of LHCII trimers. The cryo-EM map is of high quality and exhibits sufficient detailed features fitting very well with the structural model. The discovery of a new chlorophyll-binding site (Chl a615) in LHCII is novel and surprising. The result expands our knowledge on the pigment-binding capacity and function of LHCII.

I have a few comments/questions for the authors as listed below.

1) Recently, the structures and binding sites of PsbR and PsbY have been revealed in the cryo-EM structure of spinach type-I PSII-LHCII megacomplex (Shan, J. et al. *Sci. Adv.*, 10, eadq9967, 2024; DOI: 10.1126/sciadv.adq99). It will be great if the authors could compare the two small subunits found in *C. ohadii* PSII with those discovered in spinach PSII and discuss their similarity and differences. Are PsbR and/or PsbY subunits also present in the PSII-LHCII supercomplex sample purified from *C. ohadii* cells grown under low light conditions?

2) In the Extended Data Fig. 8, the authors show the change of light-harvesting protein content in the thylakoid membranes of *C. ohadii* grown under different light conditions. Are there any changes in the peripheral antenna complexes associated with PSII under different light intensities? Does PSII bind more LHCII under low light than high light, or does it just undergo LHCII content change without altering the size of antennae? It will be worthwhile to compare the low-light PSII-LHCII with the high-light one by characterizing the samples through biochemical and structural approaches (negative-stain or cryo-EM) so as to figure out the major changes of PSII-LHCII related to the adaptation under different light conditions.

3) Is the C2S2M2L2 supercomplex from *Chlorella ohadii* active in evolving oxygen and reducing plastoquinone? It will be great if the author could carry out the oxygen-evolving activity assay or use other functional analysis method to check the functional state of PSII cores in the C2S2M2L2 supercomplex from *C. ohadii*. While the OEC and Mn4O5Ca cluster are intact and well resolved in the cryo-EM map, there is a plastoquinone/plastoquinole molecule found in the QB site (the density is fairly good), suggesting that it might be trapped at the QB site and could not leave efficiently after being reduced. Is this observation related to the regulation of PSII activity in *C. ohadii*?

4) Overall, the structural model refined against the cryo-EM map has fairly good geometries for most parts. Nevertheless, some of the chlorophylls in LHCII might have a bit too large coordination bond length between Mg and its axial ligand, such as CHL 601 and 605 of chain G, CHL302 & 306, CLA315 of chain N, CLA303, CLA315 of chain R, CHL302, CLA306, CLA310, CLA316 of chain S, etc. According to the previous high-resolution structures of LHCII (PDB, 1RWT), PSI (PDB, 1JB0), PSII (PDB, 3WU2) and others, the Mg-ligand coordination usually have a bond length at 2.1-2.3 Å. It might be better to include the data of ideal distances between the coordinating pairs in the .eff file for phenix.real_space_refine program as the restraining parameters during refinement.

Minor points

1) There are five different monomers (Monomers 1-5) found in the LHCII antennae of *C. ohadii* PSII-LHCII supercomplex. What are the phylogenetic relationships between these monomers and LhcbM1-10 from *C. reinhardtii* or Lhcb1-3 from *A. thaliana*?

2) As the discovery of Chl a615 in LHCII is a highlight of this work, a more detailed description about the binding site of this chlorophyll and how it interacts with the surrounding amino acid residues might be necessary. What is the potential axial ligand of this chlorophyll, and is the C-terminal motif involved in binding Chl a615 conserved in other green algae and plants?

Reviewer #4

(Remarks to the Author)

In the context of polyamine studies, this work highlights a correlation between high light (HL) conditions and increased polyamine levels associated with thylakoid membranes. However, the data presentation in the table is somewhat difficult to read. A graph representation may be helpful. The term "sum of polyamines" appears to refer to the sum of free, bound, and conjugated polyamines.

Elevated polyamine levels are commonly observed in stress-tolerant species. Some questions arise from these findings, and I hope they will be helpful to better define a potential role of polyamines in phototolerance in this species.

- What are the bulk polyamine levels in the cell under normal light (NL) and HL conditions?
- What percentage of the bulk polyamines are associated with the thylakoid membranes of *Chlorella ohadii* under HL and NL conditions?
- Is there a specific enrichment of polyamines in thylakoid membranes under HL conditions?
- Are polyamine levels in the chloroplast higher than in cytosol?
- Is it possible that polyamines help scavenging ROS in the chloroplast under HL conditions as a mechanism to improve tolerance?

Reviewer #5

(Remarks to the Author)

General

In their paper, Arshad and colleagues explore the Cryo-EM structure of photosystem II supercomplex of the HL resistant alga *Chlorella ohadii*. A 2.9 Å resolution structure is combined with spectroscopic studies and profiling of amine compounds. Through most of the paper, results are compared to the model green algae *C. reinhardtii* on which ample relevant data is available. This work is the second Cryo-EM based structure of this PSII supercomplex with similar resolution. While several insights are provided by this version, including the identification of the specific Lhcb proteins that link different LHCs to the core complex and their light dependent expression, and several modifications of the structure (the presence of the PsbQ subunit here in the luminal side compared with PsbU in Fadeeva et al., different residues stabilizing PQ in the Qb pocket, etc.), much of the structural properties has already been revealed (see several examples below) limiting the impact of this work. I leave it to the editors to decide how incremental is the new work, but regardless of this decision, several concerns need to be addressed.

First, in many cases a thorough discussion of differences compared with Fadeeva et al is missing. Just as one example, but see many more below - How can two models with similar resolution identify different subunits in the luminal sides (PsbQ vs. PsbU)? Can this be explained by growth conditions? Are there additional factors in the experimental design that can affect?

In addition – the authors generated structures under three very different light levels, but for most cases do not specify what was the difference between them regarding specific aspects of the structure which are discussed. With or without connection to the previous point, this must be specified. Another point to consider here is the use of High CO₂ levels in this work (why??). The authors should provide a justification for this in an organism which was not isolated from a carbon rich environment.

Detailed

Line 52 – While it is true that *C. ohadii* lacks the LhcSR3 gene, and does not exhibit classical NPQ, the fact is that psbS is present and expressed in HL transcriptome of this alga, so for the sake of accuracy I would remove from this gene from the statement or at least qualify it.

Results – please try to explain or at least discuss the differences in the luminal side (here PsbO, PsbQ and PsbP) compared with Fadeeva et al (PsbO, PsbU and PsbP), especially the replacement of PsbQ and PsbU.

Lines 155-164 – the unique PsbO variant is indeed an interesting point with major potential effect on the stability, but some details are missing regarding the analysis presented in Ext. Fig. 4, e.g. Are the structural models presented based on sequence prediction (alphafold, other) or are they part of the or where they obtained from this structural work? Were the sequences validated in some way? Under which of the conditions here was the O₂ variant found in the structure, and can this be an explanation to the 8BD3 from the previous structure? This is plausible as PsbO₂ has been previously implicated

with D1 turnover (Lundin et al).

Lines 165-181 – the stronger PQ binding in the Qb pockets, including the potential tight interaction with the isoprenoid tail. It would be useful to mention this agreement here, and provide the reader with this work interpretation of the mechanism (hydrogen bonds (H-bonds) with His215 and Phe265, D2-Leu43) compared with the Figure 4 in Fadeeva et al (where at least Phe265 is presented, and other residues from D2). Again, can this be related to the differences in the conditions? Were the interactions in Ext. Fig. 5 here predicted from NL, HL, SHL or reproduced in all? In addition, cultures in this study were grown under C rich conditions (TAP and 2% CO₂, can this be a major driver of the differences here (see also the authors comments on lines 425-428)? In this context, it should be highlighted here that C supply from ambient air is likely to be more representative to desert crusts than 2% CO₂, so this aspect should at least be discussed, if not examined using air-levels CO₂ cultures.

PsbR(Psb10)/PsbY – this is yet another example where the position or appearance of subunits is reported in both structures of Ohadii, with some differences in the interactions, and therefore can support different proposed roles. As above, readers of this work would benefit from more context (i.e. growth conditions, experimental setup) on the differences observed and whether they can resolve the role suggested here compared with the recently published structure (e.g. stabilising stacked PSII, additional shielding of b559).

Lines 425-428 – there is a major confusion here by the authors. That *C. reinhardtii* antenna response is also modulated by C supply, does not mean light is not playing a (major!) role. Both factors are simply the donor and acceptor side of the same electric line called linear electron flow of photosynthesis. Even in cases where some reports claim C levels can be a factor of its own (Ruiz-Sola et al. 2023), there is clearly a role for light levels under any given C level (see e.g. Figures 1, 2, 3 and many more). Similarly, *C. ohadii* antennae response cannot be claimed to be primarily driven by light when only this factor was tested. To test the effects of C levels on *C. ohadii* antennae response, the authors should have examine different C levels on the same illumination level, or at the very least compare the antennae structure here to Fadeeva et al, where air-level CO₂ was provided to the cultures (assuming acetate is no longer available in any of the cases). There are many aspects in which *C. ohadii* differs from *C. reinhardtii*, but photosynthesis in both organisms will necessarily respond to both ends of this chain. Responding to one of them, does not eliminate the response to the other.

Polyamines – while the observation regarding the significant increase in polyamine levels is indeed novel and interesting, I find it too preliminary to be able to extract such a detailed model of their action on PMF, which is highly speculative. To better support that, similar measurements should also be done on *C. reinhardtii* under different illuminations, in addition to testing pH and other factors in *C. ohadii* upon inhibition of polyamine synthesis and/or supplementation thereof.

Version 1:

Reviewer comments:

Reviewer #1

(Remarks to the Author)

Dear Authors,

Thank you for providing detailed, point-by-point responses to all my questions and comments, and introducing changes and corrections to results of the spectroscopic analysis, all according to my suggestions. I am fully satisfied with those. I have no additional comments or suggestions.

Sincerely,

Dariusz M. Niedzwiedzki

Reviewer #2

(Remarks to the Author)

My comments were addressed and I have no further comments

Reviewer #3

(Remarks to the Author)

The manuscript has been improved after revision. The authors have addressed most of my previous questions/comments in a constructive or reasonable way. Nevertheless, there are two minor points remaining to be addressed further.

1) The following is a list of chlorophyll-ligand pairs in the peripheral antennae (L-LHCII and M-LHCII) with coordination bond lengths having fairly large errors.

Chains 11&14, 12&15 and 13&16: CLA602 and Glu 79, CLA603 and His82, CHL609 and Glu155, CLA610 and Glu197, CLA611 and LHG615, CLA612 and Asn200, CLA613 and Gln214, CLA614 and His229.

Chains 1&4: CHL609 and Glu155, CLA610 and Glu197.

Chains 2&5: CLA610 and Glu197.

It is likely that the .eff file the authors used for structure refinement does not include the coordination bond length parameters for the above pairs as restraints during the refinement process. The errors should be fixed as much as possible by adding

and applying the geometry restraints during the refinement process. An example for the coordination bond length restraint to be added in the .eff file is provided below for the author's reference.

```
geometry_restraints {
  edits {
    bond {
      atom_selection_1 = chain 11 and resseq 79 and name OE2
      atom_selection_2 = chain 11 and resseq 602 and name MG
      distance_ideal = 2.2
      sigma = 0.05
    }
  }
}
```

Besides, some chlorophyll molecules have Mg atoms protruding away from (instead of toward) the ligand, such as CLA615 of chains y&Y. The wrong configuration should also be fixed too.

2) In the newly added ED Fig. 6, the amino acid sequences of *Chlorella ohadii* PsbR and PsbY do not appear to align well with those of spinach PsbR and PsbY at first sight. Do the period symbols (.) stand for the amino acid residues identical to those of *C. ohadii* PsbR? If that is the case, please explain the meanings of the period symbols and dash symbols (-) in the legends of ED Fig. 6 and the other ED figures with sequence alignment data (ED Figs. 4b, 5c, 8, 10) to avoid confusion.

Reviewer #4

(Remarks to the Author)

In the new version of the manuscript, the authors have improved the presentation of polyamine levels, which were somewhat confusing in the previous version. The interpretation of the polyamine data is now more careful and less speculative. I agree that this approach is appropriate for the manuscript's message, and that a more detailed study of the role of polyamines would constitute a separate piece of work. The authors have addressed most of my comments and provided convincing arguments for the data that were not included.

Reviewer #5

(Remarks to the Author)

I find the revised manuscript to be largely improved. First, the added value and comparative aspects with Fadeeva et al. are now more clearly presented, with the new insights well explained to the readers. In addition, the comparison to spinach adds an important dimension to the discussion of what may contribute to *C. ohadii* extreme resistance. Finally, some (over)statements have now been qualified to be more precise and careful. I have no additional major points to raise on this work.

Original title: **Cryo-EM structure of photosystem II supercomplex from *Chlorella ohadii*, a green microalga with extreme phototolerance**

Reply to the reviewers' comments

Reviewer 1:

We would like to thank the reviewer for his/her positive evaluation and suggestions for further improvement of our manuscript. Here are our responses to the comments raised.

Page 3, lines 86-91: "This structure reveals several novel features that may hold the key to the exceptional robustness of this organism in challenging environments, including high light intensity. In particular, the structure reveals the presence of two additional PSII subunits, PsbR and PsbY, which have not been observed in previously published PSII structures of other organisms. Our findings, which are consistent with a recent study²⁴, suggest that the PsbR and PsbY subunits contribute significantly to the rigidity of the PSII complex."

Page 3, lines 116-118:" The *C. ohadii* PSII structure showed the presence of two additional subunits PsbR and PsbY (Figs. 1,2), which have not been identified in the previously characterized PSII structures from plants and other algae."

Very recently PsbR and PsbY subunits were resolved in cryo-EM structure of C₄S₄M₂ PSII-LHCII and are published (Shan, J.; Niedzwiedzki, D. M.; Tomar, R. S.; Liu, Z.; Liu, H., Architecture and functional regulation of a plant PSII-LHCII megacomplex. *Sci Adv* 2024, 10, eadq9967, DOI: 10.1126/sciadv.adq9967).

Abovementioned parts of the manuscript should be adequately amended. Moreover, since the referred above paper extensively elaborates on roles of PsbR and PsbY subunits, I suggest a careful reevaluation of the manuscript section titled "Detailed analysis of PsbR and PsbY subunit integration within the PSII core complex" on Pages 5-6, Lines 182-231.

Answer: We agree with the reviewer's suggestion to include the very recent paper by Shan et al. (2024) and to discuss it in context of our findings. In addition, we have included in the revised manuscript a new Extended Data Fig 6 showing a comparison of (i) structural models of PsbR and PsbY subunits in *C. ohadii* and spinach, and (ii) the amino acid sequences with highlighted amino acids involved in specific interactions with neighboring subunits. We have revised the relevant sections of the manuscript accordingly, and the changes are highlighted in bold.

... PsbR and PsbY, which have not been observed in previously published PSII structures of other organisms **until recently**. Our findings, which are consistent with a recent study²⁴ **and further supported by a newly resolved structure of a plant C₄S₄M₂ PSII megacomplex (Shan et al., 2024)**, suggest that the PsbR and PsbY subunits contribute significantly to the rigidity of the PSII complex **and PSII megacomplex**. (lines 100-103, the line numbers are valid for the file with tracked changes)

... The *C. ohadii* PSII structure **revealed** the presence of two additional subunits, PsbR and PsbY (Figs. 1,2), which **had** not been **resolved** in previously characterized PSII structures from plants and other algae, **but were recently observed in the spinach C₄S₄M₂ PSII-LHCII megacomplex structure (Shan et al., 2024)**. (lines 128-131)

... Several studies have suggested considerably diverse roles for PsbR, which include: stabilization of PSII complex, functioning of OEC, role in the PSII repair cycles, binding of stress-related proteins, and formation of stacked PSII32–35. **Recently, Shan et al. (2024) resolved the PsbR and PsbY subunits in**

the C4S4M2 PSII-LHCII megacomplex from spinach, where they were shown to mediate dimerization of adjacent PSII-LHCII supercomplexes in the thylakoid membrane plane. (lines 217-220)

... On the luminal side, the C-terminal loop of PsbR interacts only with the N-terminus of the PsbP subunit (Extended Data Table 3). **These interactions are mostly identical to those observed in the recent PSII structure of *C. ohadii* by Fadeeva et al. (2023). However, when compared with those described for spinach PSII-LHCII³⁷, they appear more extensive, as in spinach PsbR primarily contacts PsbC, PsbD, and PsbJ and forms only a limited number of interactions (Extended Data Fig. 6). (lines 233-237)**

... This contrasts with other green algae and land plant representatives where PsbR is not as tightly integrated into PSII supercomplexes, presumably due to weaker binding interactions. **This observation is consistent with the recent structure of the spinach PSII-LHCII megacomplex, in which PsbR and PsbY were identified only at the dimer interface between two adjacent PSII-LHCII supercomplexes. At this interface, interactions between adjacent supercomplexes appear to locally stabilize the binding of the two subunits. In contrast, on the opposite sides of the PSII-LHCII supercomplexes, which are exposed to the external environment and lack such stabilizing contacts, PsbR and PsbY were not resolved, likely due to their weak binding and subsequent loss during the isolation procedure (Shan et al. 2024). (lines 256-262)**

... These interactions suggest a multifaceted role for PsbY in stabilizing the PSII complex in *C. ohadii*, an aspect not previously observed to such an extent in cyanobacteria. **Interestingly, in spinach PSII-LHCII megacomplexes, PsbY also forms similar stabilizing interactions involving Arg and Gln residues with PsbF and PsbE (Shan et al., 2024) (Extended Data Fig. 6), supporting their conserved structural role across some plant species. (lines 274-278)**

Steady-state spectroscopy

Page 15, lines 501-503: "Absorption spectra were recorded on a Varian Cary 4000 UV-VIS spectrophotometer. Emission spectra were recorded on a HORIBA JobinYvon-Spex Fluorolog 3.22 spectrofluorimeter at an optical density of <0.05 cm⁻¹ at the Qy maximum". Considering what was written in this section of the manuscript, "optical density" value at Qy band of Chl a was directly used to calculate sample concentration (Chls content). Therefore, it assumes it is scattering free sample. Thus, more appropriate will be to use term Absorbance instead of Optical Density. The latter one incorporates also sample reflectance and scattering which as it sounds are negligible here.

Answer: We thank to the reviewer for pointing out this inaccuracy. In the revised version of the manuscript, the term "optical density" was changed to "absorbance".

Page 11: Figure 4 a and b: Absorption and Emission are physical processes and such names should be avoided on Y-axis to describe strength of the emission or absorption signal. I suggest changing them to "Fluorescence intensity" and "Absorbance", respectively.

Answer: We have changed the description of the Y-axis accordingly.

Transient absorption spectroscopy

Page 15 line 524: "Power studies were performed to make sure that annihilation effects were absent in the measurements."

The sentence mentioned that these studies were done but results are nowhere to be found to validate them. Provide (in SI) a graph with exemplary TA trace probed at some specific wavelength (like 680 nm or so) for few different excitation powers/intensities. Normalize them at maximum to show that dynamics are not affected and overlap.

Answer: These graphs have been added to the SI. For the measurement presented in the manuscript an excitation energy of 1.5 nJ and 0.6 nJ (at a spot size of 150 μm) was used for respectively the *C. ohadii* and *A. thaliana* LHCII trimers, showing no indications of annihilation at these excitation powers.

Moreover, there is no information what excitation power (or energy, or fluence, and excitation beam size on the sample) was used to obtain results presented in this work.

Answer: These details have been added to the methods section: **“The pump–and-probe pulse had FWHM’s of respectively ~150 μm and ~75 μm . The samples were measured in a 1 mm cuvette that was constantly shaken throughout the measurements. Power studies were performed to make sure that annihilation effects were absent in the measurements (see Extended Data Fig. 14). For the measurement presented in the manuscript an excitation energy of 1.5 nJ and 0.6 nJ at a spot size of 150 μm was used for respectively the *C. ohadii* and *A. thaliana* LHCII trimers.”** (lines 607-611)

Transient absorption of LHCII trimers

Page 10-11, lines 366-381, Figure 4C:

I do understand motivation for comparison of TA of LHCII from both species. However, it should be also discussed in perspective to what is known on this subject from past studies. This is not a novel subject. There is a vast number of papers on it and not even one was referred here.

Answer: The following sentence has been added that acknowledges the vast amount of ultrafast spectroscopic studies on LHCII from the past:

“Numerous time-resolved spectroscopy studies have been conducted on LHCII in the past, e.g.¹⁻⁴ and the presented data here for both *C. ohadii* and *A. thaliana* is in line with them, showing fast inter-Chl excitation energy equilibration followed by a slow excited state decay. Little attention has however been devoted to the comparison of the ultrafast excitation energy pathways of LHCII across species, with the exception of a few studies^{5,6}.” (lines 439-443)

Why last two DADS were taken out of the analysis shown in Figure 4c? It is convenient but readers would like to know what they represent.

Answer: As we focus on the consequences of the extra Chl on the EET dynamics we have opted not to show the DADS that describe the excited state decay of the system (the last two DADS) in the main figure, but we do show them in the SI. A sentence describing them has been added to the text: **“The last two DADS for both complexes (Extended Data Fig. 12) describe the decay of the Chl excited states, which in a small fraction of the complexes proceeds on the order of 150 ps, but for most of them takes relatively long (> 3.5 ns), reflecting the mixture of respectively quenched and unquenched complexes in the ensemble, which is typical of those systems in detergent micelles⁷.”** (lines 454-458)

Based on TA data description, discussion and methodology one would assume that authors present annihilation free results. I am not so convinced. The 150-200 fs and 2-3 ps DADS are indeed associated with EET process within monomer (positive-negative shape) however 130-170 ps and ~4 ns DADS have the same (only negative) profiles of very different amplitudes. It means these are mutually independent but share the common spectral shape. Since 4 ns DADS is associated with intrinsic decay of Chl a excited singlet state what 130-170 ps DADS does represent? In my opinion it is single-singlet annihilation on monomer-monomer LHCII interface within the trimer.

Answer: The power studies indicate that the data are annihilation free, amplitude of the decay of the 130-170 ps component accounts for 20% of the total signal. Moreover, singlet-singlet annihilation in LHCII trimers mainly occurs with a 20-30 ps lifetime^{3,8-10} and not 150 ps. The 150 ps component is due to a quenched conformation of LHCII in solution as it is now also mentioned in the manuscript (see above).

I think all these will play out and will be much better seen if instead of the most basic parallel excitation decay model a more intuitive sequential scheme is used for TA fitting. I am sure it could be

easily implemented in pyglotaran. Both models produce macroscopic rates (so lifetimes do not change) but the sequence gives spectral evolution of the bleaching of Qy band of Chls. Such presentation will be substantially more digestible for readers who may clearly see resemblance of Qy band absorption and its bleaching and how it changes over delay time from moment of sample excitation. This is not so easy to decipher from meaningless shapes of DADS profiles. For visualization I encourage authors to look into this paper (Gradinaru et al. Selective Interaction between Xanthophylls and Chlorophylls in LHCII Probed by Femtosecond Transient Absorption Spectroscopy, <https://doi.org/10.1021/jp026278q>) and particularly focus on Figure 2B and accompanied discussion on singlet-singlet annihilation.

Answer: The EADS have been added to the SI for the interested readers. We believe that the DADS are more illustrative to show excitation energy transfer processes as they are nicely interpretable as flowing from the negative amplitudes to the positive ones, and allow for a better comparison.

Is 671 nm excitation beam profile sufficiently narrow to excite only Chls a in the LHCII trimer or it will also partially overlap with Chl b absorption? Could authors add one more figure in the SI that shows both, Qy band absorption of LHCII and laser pulse spectrum, so there will be clarity about it.

Answer: We have added the pulse spectrum to the SI (Extended Data Fig. 11).

Reviewer 2:

We would like to thank the reviewer for his/her positive evaluation and suggestions for further improvement of our manuscript. Here are our responses to the comments raised.

Key structural findings apart from the discovery of a unique PsbO isoform, are not entirely novel, as a very similar structure of the photosystem II C2S2M2L2 supercomplex from *C. ohadii* was already published by Fadeeva et al. (2023, PDB 8BD3, reference 24 in the current article). The roles of subunits PsbR (Psb10) and PsbY as photoprotectors, as well as their stabilizing interactions with cytochrome b559, were previously proposed. The additional chlorophyll-binding site (Chl a615) in LHCII was also present in the earlier 8BD3 structure. It is concerning that readers might sense the authors are attempting to obscure the previous publication rather than clearly stating the differences between the two structures. The authors need to explicitly highlight how their results represent clear advancements or novel insights beyond those previously reported, specifically addressing comparisons with Fadeeva et al.

Answer: We thank the reviewer for raising this important point. We have carefully revised the manuscript and have aimed to objectively compare and discuss our findings in relation to those reported by Fadeeva et al. (2023) throughout the text. However, it should be noted that the mentioned paper does not include description of several interesting structural aspects of the *C. ohadii* PSII supercomplex. For example, there is no detailed description of the interactions between the subunits, no mention of the presence of Chl a615 and its binding environment within the LHCII trimers, or the specific composition of the LHCII trimers. These structural features are only evident upon a thorough analysis of the deposited structural model, as they are not described or discussed in the text of the publication. In addition to these omissions, the study contains some inconsistencies that may be confusing to readers. For example, the OEC subunit PsbU was mislabeled in their structure because its location and structure correspond to PsbQ, a known OEC component in green algae and plants, whereas PsbU is a cyanobacterial subunit with a different structure and binding site. Similarly, the reported total number of pigments in the text does not match the content of the structural model. We believe that the structural interpretation and biological context provided in our manuscript offer clearer insights and may be of greater relevance to the scientific community than if readers were left to deduce these aspects independently from the structural file.

In addition, in the revised version of the manuscript the structural role of PsbR and PsbY is discussed in the context of the recent structure of PSII-LHCII megacomplexes from spinach, where both subunits were also revealed (Shan et al. 2024). In the revised version of the manuscript, we have included a new Extended Data Fig 6 showing a comparison of (i) structural models of PsbR and PsbY subunits in *C. ohadii* and spinach, and (ii) the amino acid sequences with highlighted amino acids involved in specific interactions with neighboring subunits.

In the revised version of the manuscript, the main changes to the text reflecting the above points are as follows (the line numbers are valid for the file with tracked changes):

- acknowledgements of articles by Fadeeva et al. (2023) and Shan et al. (2024) in Introduction part (lines 88-103)
- pointing out the mislabeling of the OEC subunit PsbU instead of PsbQ (lines 168-171)
- detailed description of the binding position of PQ in the QB pocket and interactions involved in the stabilization of the PQ molecule (lines 191-196)
- the chapter "Detailed analysis of PsbR and PsbY subunit integration within the PSII core complex" was revised with respect to papers by Fadeeva et al. (2023) and Shan et al. (2024) (lines 210-278)
- the interactions and binding positions of Chl a615 was described in more detail and differences from the structural model of Fadeeva et al. (2023) are discussed (lines 304-324)

Additionally, specific Lhcb proteins within the antenna complexes were found to modulate the binding of light-harvesting antenna trimers in response to light conditions, supported by mass spectrometry data showing downregulation under high-light conditions. The study also found a significant increase

in polyamines in high-light acclimated samples, suggesting a photoprotective role. Collectively, the results suggest a distinct photoprotection strategy in *C. ohadii*, relying on a rigid PSII core, specialized antenna composition, and polyamine-regulated electron transport and stability of the thylakoid membrane.

Although the structure was presented as the primary result of the manuscript, the manuscript's real strength lies in additional analyses such as mass spectrometry, spectroscopy, LC-MS/MS, RNA-seq, and FRET, particularly given that Fadeeva et al.'s publication relied solely on structural data. Hence, the focus of the manuscript might need to be revised by the authors.

Answer: As recommended, in the revised manuscript we emphasized the novel findings in comparison with the results reported by Fadeeva et al. (see our response above). As to the results of the measurement of polyamines in thylakoid membranes isolated from NL and HL *C. ohadii* cells, we added two figures into the manuscript (Extended Data Fig. 13a and b) based on the data presented in Extended Data Table 7. The new figures clearly shows that the total polyamine content increased more than 10 times in HL when compared to NL thylakoid membranes and also which individual polyamines are related to this phenomenon. High accumulation of specific polyamines is also indicated in the revised abstract of our manuscript. However, due to concerns raised by other reviewers that our proposed role of polyamines in regulation of electron transport is not supported by additional experimental data, we decided to revise the text of the last chapter "What factors contribute to the high-light resistance of *Chlorella ohadii*?" (lines 495-538).

Line 74- The sentence discussing LHCII trimers should cite algae PSII structures and not only plant structures.

Answer: We have added the references to PSII structures from green alga *C. reinhardtii* (Tokutsu et al. (2012) JBC; Drop et al. (2014) BBA; Sheng et al. (2019) Nat Plants; Shen et al. (2019) PNAS).

Line 82 - The sentence referring to cryo-EM structures of *C. reinhardtii* should mention all known PSII structures from algae, including *Dunaliella* and *Chlorella*.

Answer: In the revised manuscript, we have added citations to papers concerning cryo-EM structures of PSII from various algal species, including *Dunaliella*, *Chlorella*, diatoms, cryptophytes and haptophytes. (lines 87-88)

Line 86 -90. The authors claim novel features that have not been observed previously, yet this overlaps significantly with Fadeeva et al. (2023).

Answer: We have modified the last two paragraphs of the introduction to mention the overlap with the study by Fadeeva et al. (2023) and to outline how our work extends and complements their findings.

Line 302 -suggested editing: In contrast, *C. ohadii* employs different photoprotective mechanisms, as neither LhcSR nor NPQ is involved.

Answer: We have modified the sentence according to the reviewer's suggestion.

Line 422 - see Extended Data Fig. 8 and citations of papers by Isreal groups"—this should explicitly refer to specific citations.

Answer: We have indicated specific citations in the revised manuscript.

Line 435 – suggested editing: However, polyamine levels increased more than tenfold in the HL-acclimated sample compared to the NL sample.

Answer: We have modified the sentence according to the reviewer's suggestion.

References 24 and 38 are duplicates and need correction.

Answer: We thank the reviewer for pointing out this error. We have corrected it.

What was the reason for comparing the spectroscopic data to Arabidopsis rather than to a more closely related alga, such as Chlamydomonas?

Answer: The pigment number, composition and structural organization of LHCII are essentially the same in both species. However, the LHCII complex from Arabidopsis is the best characterized by far in terms of structure and spectroscopy, making it the best reference point for comparison.

Likewise, what was the reason for comparing PsbO to diatom and red algae? I suggest adding sequences from the green lineage.

Answer: We thank the reviewer for this comment. In our initial analysis, we screened available PSII structures from a range of photosynthetic organisms to examine the type of PsbO isoform resolved in each case. In the revised manuscript, we have updated Extended Data Fig. 4 to include the structure of PsbO from *Pisum sativum*, a representative of angiosperms, to better reflect the structural variation within the green lineage.

The discussion lacks a clear connection between specific structural features and photoprotection.

Answer: We thank the reviewer for this important comment. In our study, we relate the higher stability of the PSII structure from *C. ohadii* primarily to the stable binding of the PsbR and PsbY subunits, which likely contribute to the overall rigidity and stabilization of the PSII core complex. The observed antenna size regulation under high-light (HL) conditions, based on our MS analysis, is consistent with previous findings by Levin et al., and is now more clearly described in the final chapter titled “What factors contribute to the high-light resistance of *Chlorella ohadii*?”. We also propose a possible role of increased polyamine levels under HL in contributing to the high phototolerance of *C. ohadii*.

However, we agree that a direct correlation between structural features and photoprotection would require structural data from PSII supercomplexes isolated under HL conditions. While the reduction of large PSII C₂S₂M₂L₂ supercomplexes to smaller forms such as C₂S₂ or even C₂ is suggested by previous biochemical studies, potential structural changes within the core complex itself may play an important role in the exceptional phototolerance of *C. ohadii*. Determining such changes is beyond the scope of the current study, but we are actively working on resolving the PSII structure from HL-grown *C. ohadii*, as we consider it highly relevant to further elucidate the mechanisms of PSII photoprotection in this unique organism.

Discrepancies exist between chlorophyll numbers mentioned in the text and those found in the structural model—for example, chlorophyll 615 of the PsbN subunit is absent from the provided model cif file.

Answer: We thank the reviewer for pointing out this discrepancy. Based on this comment, we identified formatting errors in the “cif” structural file obtained from the PDB database compared to our originally submitted file. We have now corrected the structural file, in which the ligand IDs and associated chains have been updated accordingly.

Reviewer 3:

We would like to thank the reviewer for his/her positive evaluation and suggestions for further improvement of our manuscript. Here are our responses to the comments raised.

Recently, the structures and binding sites of PsbR and PsbY have been revealed in the cryo-EM structure of spinach type-I PSII-LHCII megacomplex (Shan, J. et al. *Sci. Adv.*, 10, eadq9967, 2024; DOI: 10.1126/sciadv.adq999). It will be great if the authors could compare the two small subunits found in *C. ohadii* PSII with those discovered in spinach PSII and discuss their similarity and differences. Are PsbR and/or PsbY subunits also present in the PSII-LHCII supercomplex sample purified from *C. ohadii* cells grown under low light conditions?

Answer: We agree with the reviewer's suggestion to include the very recent paper by Shan et al. (2024) and to discuss it in context of our findings. In addition, we have included in the revised manuscript a new Extended Data Fig 6 showing a comparison of (i) structural models of PsbR and PsbY subunits in *C. ohadii* and spinach, and (ii) the amino acid sequences with highlighted amino acids involved in specific interactions with neighboring subunits. We have revised the relevant sections of the manuscript accordingly, and the changes are highlighted in bold.

... PsbR and PsbY, which have not been observed in previously published PSII structures of other organisms **until recently**. Our findings, which are consistent with a recent study²⁴ **and further supported by a newly resolved structure of a plant C₄S₄M₂ PSII megacomplex (Shan et al., 2024)**, suggest that the PsbR and PsbY subunits contribute significantly to the rigidity of the PSII complex **and PSII megacomplex**. (lines 100-103, the line numbers are valid for the file with tracked changes)

... The *C. ohadii* PSII structure **revealed** the presence of two additional subunits, PsbR and PsbY (Figs. 1,2), which **had** not been **resolved** in previously characterized PSII structures from plants and other algae, **but were recently observed in the spinach C₄S₄M₂ PSII-LHCII megacomplex structure (Shan et al., 2024)**. (lines 128-131)

... Several studies have suggested considerably diverse roles for PsbR, which include: stabilization of PSII complex, functioning of OEC, role in the PSII repair cycles, binding of stress-related proteins, and formation of stacked PSII₃₂₋₃₅. **Recently, Shan et al. (2024) resolved the PsbR and PsbY subunits in the C₄S₄M₂ PSII-LHCII megacomplex from spinach, where they were shown to mediate dimerization of adjacent PSII-LHCII supercomplexes in the thylakoid membrane plane.** (lines 217-220)

... On the luminal side, the C-terminal loop of PsbR interacts only with the N-terminus of the PsbP subunit (Extended Data Table 3). **These interactions are mostly identical to those observed in the recent PSII structure of *C. ohadii* by Fadeeva et al. (2023). However, when compared with those described for spinach PSII-LHCII³⁷, they appear more extensive, as in spinach PsbR primarily contacts PsbC, PsbD, and PsbJ and forms only a limited number of interactions (Extended Data Fig. 6).** (lines 233-237)

... This contrasts with other green algae and land plant representatives where PsbR is not as tightly integrated into PSII supercomplexes, presumably due to weaker binding interactions. **This observation is consistent with the recent structure of the spinach PSII-LHCII megacomplex, in which PsbR and PsbY were identified only at the dimer interface between two adjacent PSII-LHCII supercomplexes. At this interface, interactions between adjacent supercomplexes appear to locally stabilize the binding of the two subunits. In contrast, on the opposite sides of the PSII-LHCII supercomplexes, which are exposed to the external environment and lack such stabilizing contacts, PsbR and PsbY were not resolved, likely due to their weak binding and subsequent loss during the isolation procedure (Shan et al. 2024).** (lines 256-262)

... These interactions suggest a multifaceted role for PsbY in stabilizing the PSII complex in *C. ohadii*, an aspect not previously observed to such an extent in cyanobacteria. **Interestingly, in spinach PSII-LHCII megacomplexes, PsbY also forms similar stabilizing interactions involving Arg and Gln residues with PsbF and PsbE (Shan et al., 2024) (Extended Data Fig. 6), supporting their conserved structural role across some plant species.** (lines 276-278)

An additional note: Our structure was solved from PSII supercomplexes isolated from *C. ohadii* cells grown under normal light (NL) conditions ($100 \mu\text{mol photons m}^{-2} \text{s}^{-1}$). We did not analyze PSII from cultures grown under low light (LL) conditions, so we cannot confirm the presence or absence of the PsbR and PsbY subunits in LL-grown cells. This remains an interesting question for future investigation.

2) In the Extended Data Fig. 8, the authors show the change of light-harvesting protein content in the thylakoid membranes of *C. ohadii* grown under different light conditions. Are there any changes in the peripheral antenna complexes associated with PSII under different light intensities? Does PSII bind more LHCII under low light than high light, or does it just undergo LHCII content change without altering the size of antennae? It will be worthwhile to compare the low-light PSII-LHCII with the high-light one by characterizing the samples through biochemical and structural approaches (negative-stain or cryo-EM) so as to figure out the major changes of PSII-LHCII related to the adaptation under different light conditions.

Answer: There are several biochemical studies (Levin et al. (2021) *Plant J* 106; Levin et al. (2023) *Plant J* 115; Levin et al. (2025) *Plant Physiology* 197; cited in our manuscript) demonstrating a significant reduction of PSII antenna size in *C. ohadii* under high-light conditions. These biochemical data indicate a gradual transition of large PSII $\text{C}_2\text{S}_2\text{M}_2\text{L}_2$ supercomplexes into smaller forms or PSII core complexes. Our current MS data support this model, showing that the light-induced reduction of specific LHCII proteins limits LHCII trimer binding to PSII. We agree that direct structural comparison of PSII supercomplexes from cells grown under different light conditions would be highly informative. This is a major goal of our ongoing research, which focuses not only on elucidating antenna size adjustments but also on potential light-dependent changes in the PSII core structure that may be missed by biochemical analyses.

3) Is the $\text{C}_2\text{S}_2\text{M}_2\text{L}_2$ supercomplex from *Chlorella ohadii* active in evolving oxygen and reducing plastoquinone? It will be great if the author could carry out the oxygen-evolving activity assay or use other functional analysis method to check the functional state of PSII cores in the $\text{C}_2\text{S}_2\text{M}_2\text{L}_2$ supercomplex from *C. ohadii*. While the OEC and Mn₄O₅Ca cluster are intact and well resolved in the cryo-EM map, there is a plastoquinone/plastoquinole molecule found in the QB site (the density is fairly good), suggesting that it might be trapped at the QB site and could not leave efficiently after being reduced. Is this observation related to the regulation of PSII activity in *C. ohadii*?

Answer: Unfortunately, the oxygen-evolving activity of the isolated PSII $\text{C}_2\text{S}_2\text{M}_2\text{L}_2$ supercomplex used for structural cryo-EM analysis was not measured, as the available material was prioritized for biochemical and structural characterization, including several test experiments. As we now better describe in the revised manuscript (lines 185–209), the observed density of the plastoquinone (PQ) molecule in the QB site is likely stabilized by specific interactions with surrounding residues forming the QB pocket, which are essential for efficient electron transfer in PSII. We included two more references to supporting this observation (Brown et al. 2024, Saito et al. 2013). In the study by Fadeeva et al. (2023), the authors speculate that the QB pocket may accommodate multiple PQ molecules, enabling reoxidation via a cascade mechanism without requiring the reduced PQ to leave the pocket. However, based on our structural comparison, we do not support this interpretation, as we did not observe a significantly enlarged QB pocket in our structure. We recognize the importance of functional validation and plan to address this aspect in our ongoing work. Specifically, we are currently focusing on the structural and functional analysis of PSII complexes isolated from high-light (HL) grown *C. ohadii* cultures. This will allow us to evaluate the photochemical activity of PSII under different light regimes

and gain deeper insight into the role of the acceptor side, particularly the QB site, in the phototolerance and electron transport capacity of this organism.

4) Overall, the structural model refined against the cryo-EM map has fairly good geometries for most parts. Nevertheless, some of the chlorophylls in LHCII might have a bit too large coordination bond length between Mg and its axial ligand, such as CHL 601 and 605 of chain G, CHL302 & 306, CLA315 of chain N, CLA303, CLA315 of chain R, CHL302, CLA306, CLA310, CLA316 of chain S, etc. According to the previous high-resolution structures of LHCII (PDB, 1RWT), PSI (PDB, 1JB0), PSII (PDB, 3WU2) and others, the Mg-ligand coordination usually have a bond length at 2.1-2.3 Å. It might be better to include the data of ideal distances between the coordinating pairs in the .eff file for phenix.real_space_refine program as the restraining parameters during refinement.

Answer: The structure was refined exactly as suggested by the reviewer, i.e. using modified eff file to restrain ideal distances for all CHL and CLA ligands. However, in case of several main-chain oxygen atom interactions with Mg (exactly as pointed by the reviewer), the distance was not set up in eff file, to reduce amount of several Ramachandran outliers, which we obtained when these interactions were included during the refinement. In our refinement, we preferred zero Ramachandran outliers as is shown in the Extended Data Table 1.

We believe that this issue is mainly linked to a lower resolution in several areas of this dataset, as with other datasets we have recently collected for different protein complex at 2.1-2.3 Å resolution (will be published elsewhere), we do not have this problem anymore.

Minor points

1) There are five different monomers (Monomers 1-5) found in the LHCII antennae of *C. ohadii* PSII-LHCII supercomplex. What are the phylogenetic relationships between these monomers and LhcbM1-10 from *C. reinhardtii* or Lhcb1-3 from *A. thaliana*?

Answer: The comparison of amino sequences in LhcbM and monomers 1-5 showed that the trimerization motif (**WYXXXR**) (Hobe et al. 1995, Biochemistry) and L18 domains (**VDPLYGGSFDPGLADD**) (DeLille et al. 2000, Proc Natl Acad Sci USA) contain several amino acid substitutions. Specifically, although the comparison shows sequence matches in the trimerization motifs in four out of five monomers, the L18 domains of the *C. ohadii* Lhcb monomers show various differences in amino acid sequences. These findings indicates that the identified five monomers (monomer 1-5) of trimeric antenna complex could not be related to the LhcbM proteins from *C. reinhardtii*. In the revised version of the manuscript, we included the following text:

“Sequence comparison of monomers 1–5 with LhcbM proteins from *C. reinhardtii* revealed notable differences in the conserved L18 domain, indicating that these monomers cannot be reliably assigned to the known LhcbM types (I–IV) (Minagawa and Takahashi 2004).” (lines 336-338)

2) As the discovery of Chl a615 in LHCII is a highlight of this work, a more detailed description about the binding site of this chlorophyll and how it interacts with the surrounding amino acid residues might be necessary. What is the potential axial ligand of this chlorophyll, and is the C-terminal motif involved in binding Chl a615 conserved in other green algae and plants, and is the C-terminal motif involved in binding Chl a615 conserved in other green algae and plants?

Answer: We thank the reviewer for pointing out this insufficient description. In the revised version of the manuscript, we added the following text describing the Chl a615 ligands:

“Based on our structural model, the Chl a615 is coordinated by specific residues at the C-terminus of the Lhcb monomers. The axial ligand appears to be a leucine residue, Leu255, Leu250, and Leu242 in monomers 3, 4, and 5, respectively, while in monomer 1, Chl a615 is coordinated by Ser242. Interestingly, we also identified the same coordinating residues in the recent PSII structure of *C. ohadii* by Fadeeva et al.³⁶, where, due to the better-resolved C-termini in M- and L-LHCII chains, two Chl a615 molecules per each LHCII trimer were resolved. Sequence analysis of the C-termini of the Lhcb proteins that form the S-, M-, and L-LHCII trimers in our structural model shows that specific motifs, such as NLA, SLA, or SAGIGY (coordinating residues in bold), are important for Chl a615

binding. In contrast, the Lhcb chain, which consistently lacked Chl a615 in our structure and also in the structure of Fadeeva et al. consistently showed a FLPLP motif at the C-terminus that occupies the space of the Chl a615 site, effectively preventing its binding. Interestingly, similar motifs, FTPSA or FTPQ, are found at the C-termini of Lhcb proteins forming S-, M-, and L-LHCII trimers in *C. reinhardtii*16, where Chl a615 was not observed. These results suggest that the specific C-terminal motif of Lhcb proteins plays a crucial role in creating a favorable environment for Chl a615 binding and coordination.” (lines 311-324)

Reviewer 4

We would like to thank the reviewer for his/her positive evaluation and suggestions for further improvement of our manuscript. Here are our responses to the comments raised.

In the context of polyamine studies, this work highlights a correlation between high light (HL) conditions and increased polyamine levels associated with thylakoid membranes. However, the data presentation in the table is somewhat difficult to read. A graph representation may be helpful. The term "sum of polyamines" appears to refer to the sum of free, bound, and conjugated polyamines.

Answer: We agree with this remark. In the revised manuscript we added the Extended Data Fig. 13 with a, total polyamine content and b, the content of individual polyamines in thylakoid membranes of *C. ohadii* grown under 100 $\mu\text{mol photons m}^{-2}\text{s}^{-1}$ (NL) and 2500 $\mu\text{mol photons m}^{-2}\text{s}^{-1}$ (HL) based on the data Extended Data Table 7. These additional figures show that i) the level of total polyamines in HL thylakoids increased approximately 10 times when compared to NL thylakoids and that ii) it appeared mainly due to the high levels of diamines cadaverine, putrescine and diaminopropane.

Elevated polyamine levels are commonly observed in stress-tolerant species. Some questions arise from these findings, and I hope they will be helpful to better define a potential role of polyamines in phototolerance in this species.

- What are the bulk polyamine levels in the cell under normal light (NL) and HL conditions?
- What percentage of the bulk polyamines are associated with the thylakoid membranes of *Chlorella ohadii* under HL and NL conditions?
- Is there a specific enrichment of polyamines in thylakoid membranes under HL conditions?
- Are polyamine levels in the chloroplast higher than in cytosol?
- Is it possible that polyamines help scavenging ROS in the chloroplast under HL conditions as a mechanism to improve tolerance?

Answer: We thank the reviewer for this comment. We acknowledge that addressing these questions would provide a more comprehensive understanding of the role of polyamines in *C. ohadii* phototolerance. However, the role of polyamines in plant cell physiology is highly complex and, in our opinion, remains under-researched.

As our manuscript focuses on PSII, which is embedded in the thylakoid membrane, we chose to limit our measurements of polyamine content to the most relevant level— thylakoid membranes.

In the revised manuscript, we have omitted our detailed mechanistic model of polyamine action in the thylakoid membranes of *C. ohadii* related to phototolerance, as it was not supported by our data (as noted by Reviewer 5). Instead, we now interpret our polyamine data within the framework proposed by Kotzabasis and coworkers, who associate elevated levels of the diamine putrescine with the adaptation of photosynthetic organisms to high light conditions (for review, see Novakoudis and Kotzabasis, 2022).

This concept includes:

(i) a direct action of polyamines on the conformation of LHCII (Navakoudis et al., 2007), (ii) a direct binding of polyamines to the PSII proteins regulating the activity of PSII reaction centers (Hamdani et al., 2011), and (iii) alterations in electrical ($\Delta\psi$) and chemical (ΔpH) components of proton motive force (Ioannidis et al., 2012; Ioannidis and Kotzabasis, 2014), which regulate photochemical and non-photochemical quenching. (lines 525-529)

Reviewer 5

We would like to thank the reviewer for his/her positive evaluation and suggestions for further improvement of our manuscript. Here are our responses to the comments raised.

First, in many cases a thorough discussion of differences compared with Fadeeva et al is missing. Just as one example, but see many more below - How can two models with similar resolution identify different subunits in the luminal sides (PsbQ vs. PsbU)? Can this be explained by growth conditions? Are there additional factors in the experimental design that can affect?

Answer: We thank the reviewer for raising this important point. We have carefully revised the manuscript and have aimed to objectively compare and discuss our findings in relation to those reported by Fadeeva et al. (2023) throughout the text. However, it should be noted that the mentioned paper does not include description of several interesting structural aspects of the *C. ohadii* PSII supercomplex. For example, there is no detailed description of the interactions between the subunits, no mention of the presence of Chl $\alpha 615$ and its binding environment within the LHCII trimers, or the specific composition of the LHCII trimers. These structural features are only evident upon a thorough analysis of the deposited structural model, as they are not described or discussed in the text of the publication.

In addition to these omissions, the study contains some inconsistencies that may be confusing to readers, as we can see also from your comment. For example, the OEC subunit PsbU was mislabeled in their structure because its location and structure correspond to PsbQ, a known OEC component in green algae and plants, whereas PsbU is a cyanobacterial subunit with a different structure and binding site. Similarly, the reported total number of pigments in the text does not match the content of the structural model.

We believe that the structural interpretation and biological context provided in our manuscript offer clearer insights and may be of greater relevance to the scientific community than if readers were left to deduce these aspects independently from the structural file.

In addition, in the revised version of the manuscript the structural role of PsbR and PsbY is discussed in the context of the recent structure of PSII-LHCII megacomplexes from spinach, where both subunits were also revealed (Shan et al. 2024). In the revised version of the manuscript, we have included a new Extended Data Fig 6 showing a comparison of (i) structural models of PsbR and PsbY subunits in *C. ohadii* and spinach, and (ii) the amino acid sequences with highlighted amino acids involved in specific interactions with neighboring subunits.

In the revised version of the manuscript, the main changes to the text reflecting the above points are as follows (the line numbers are valid for the file with tracked changes):

- acknowledgements of articles by Fadeeva et al. (2023) and Shan et al. (2024) in Introduction part (lines 88-103)
- pointing out the mislabeling of the OEC subunit PsbU instead of PsbQ (lines 168-171)
- detailed description of the binding position of PQ in the QB pocket and interactions involved in the stabilization of the PQ molecule (lines 191-196)
- the chapter "Detailed analysis of PsbR and PsbY subunit integration within the PSII core complex" was revised with respect to papers by Fadeeva et al. (2023) and Shan et al. (2024) (lines 210-278)
- the interactions and binding positions of Chl $\alpha 615$ was described in more detail and differences from the structural model of Fadeeva et al. (2023) are discussed (lines 304-324)

In addition – the authors generated structures under three very different light levels, but for most cases do not specify what was the difference between them regarding specific aspects of the structure which are discussed. With or without connection to the previous point, this must be specified.

Answer: We thank the reviewer for pointing out this ambiguity. In the revised manuscript, we now clearly state that the cryo-EM structure of the PSII $C_2S_2M_2L_2$ supercomplex was solved exclusively from *C. ohadii* cells grown under normal light (NL) conditions ($100 \mu\text{mol photons m}^{-2} \text{s}^{-1}$). Cultures grown under high-light (HL, $1700 \mu\text{mol photons m}^{-2} \text{s}^{-1}$) and super-high-light (SHL, $2500 \mu\text{mol photons m}^{-2}$

s⁻¹) conditions were used only for mass spectrometry analysis to investigate light-dependent changes in light-harvesting protein abundance.

This is the revised text:

“The PSII supercomplexes were isolated and purified from *C. ohadii* grown under normal-light conditions (100 μmol photons m⁻² s⁻¹) by sucrose density gradient (SDG) ultracentrifugation.” (lines 121-122)

Another point to consider here is the use of High CO₂ levels in this work (why??). The authors should provide a justification for this in an organism which was not isolated from a carbon rich environment.

Answer: Regarding the use of high CO₂ concentrations, this approach was adopted to accelerate the growth of dense algal cultures (OD ~0.7) in a photobioreactor device, which was crucial for isolating sufficient quantities of the intact supercomplex for cryo-EM and other analyses. During photosynthesis, algal uptake of CO₂ causes a gradual depletion of dissolved CO₂, which leads to an increase in culture pH. At elevated pH, CO₂ is converted to carbonate (CO₃²⁻), a form of inorganic carbon that is less bioavailable to algae (Sutherland et al., 2015, Water Res.). To counteract this effect, additional CO₂ was supplied to maintain the pH within a physiological range and to sustain carbon availability, thereby promoting faster growth. This approach is particularly relevant for *Chlorella ohadii*, a fast-growing alga that may possess an efficient carbon-concentrating mechanism (CCM), allowing it to thrive even in its natural carbon-limited environments. Additionally, extending culture duration in a photobioreactor increases the risk of contamination. Therefore, supplementing CO₂ not only supported optimal growth under controlled conditions but also reduced the overall culture time and minimized contamination risk.

In the revised version of the manuscript, we included the following text in the Methods (Purification of *C. ohadii* PSII supercomplexes) to justify the use of high CO₂ level: **“The use of elevated levels of CO₂ was advantageous for our structural and biochemical analyses of the PSII supercomplex in *C. ohadii* because it allowed us to achieve high cell yields in the smaller culture volume of the photobioreactor (1 L), which was crucial for isolating sufficient amounts of intact supercomplex for cryo-EM and other analyses.”** (lines 545-549)

Line 52 – While it is true that *C. ohadii* lacks the LhcSR3 gene, and does not exhibit classical NPQ, the fact is that psbS is present and expressed in HL transcriptome of this alga, so for the sake of accuracy I would remove from this gene from the statement or at least qualify it.

Answer: We thank the reviewer for pointing out this inaccuracy. In the revised manuscript, we have stated the following:

“It does not possess the LhcSR gene and does not exhibit the typical non-photochemical quenching (NPQ) induction pattern. Although the PsbS gene is present and upregulated under high light conditions, its role in photoprotection remains unclear, especially when compared to its well-established function in NPQ in land plants and other green algae⁴⁻⁸.” (lines 54-59)

Results – please try to explain or at least discuss the differences in the luminal side (here PsbO, PsbQ and PsbP) compared with Fadeeva et al (PsbO, PsbU and PsbP), especially the replacement of PsbQ and PsbU.

Answer: Please see our response on this topic above.

Lines 155-164 – the unique PsbO variant is indeed an interesting point with major potential effect on the stability, but some details are missing regarding the analysis presented in Ext. Fig. 4, e.g. Are the structural models presented based on sequence prediction (alphafold, other) or are they part of the or where they obtained from this structural work? Were the sequences validated in some way? Under which of the conditions here was the O2 variant found in the structure, and can this be an explanation to the 8BD3 from the previous structure? This is plausible as PsbO2 has been previously implicated with D1 turnover (Lundin et al).

Answer: In Extended Data Fig. 4, we present a structural comparison of PsbO proteins from various organisms. All models shown in this figure were obtained from cryo-EM structures (not from AlphaFold models). The corresponding accession codes are provided in the figure legend, so we believe that the origin of the PsbO structures will be obvious to the reader. The sequences were taken directly from the deposited cryo-EM structural models, which undergo validation during submission to the Protein Data Bank, ensuring their accuracy.

As we clarified in the revised manuscript (see also our response above), the cryo-EM structure of PSII supercomplex was obtained from *C. ohadii* grown under normal-light conditions (100 $\mu\text{mol photons m}^{-2} \text{s}^{-1}$). These light conditions are comparable to those reported in the study by Fadeeva et al. (2023). The fact that the PsbO2 isoform was not reported in their structure (PDB: 8BD3) indicates that other cultivation factors are involved, e.g. CO₂ availability or aeration regime, which may affect protein expression or complex assembly.

Regarding the PsbO2, indeed, prior work has implicated PsbO2 in the PSII repair cycle, particularly in the context of D1 turnover. In *Arabidopsis*, PsbO2-deficient mutants showed impaired degradation of the D1 protein and increased sensitivity to high-light stress, while PsbO2 also exhibited higher GTPase activity compared to PsbO1, suggesting its regulatory role in light-induced, GTP-dependent D1 protein degradation (Lundin et al. 2008; Spetea & Lundin 2012). This supports the notion that PsbO2 contributes to PSII maintenance and repair, a role that may be relevant in *C. ohadii* as well, given our structural findings.

In the revised version of the manuscript, we included the following text:

“It should be noted that land plants such as *Arabidopsis thaliana* also encode two PsbO isoforms, PsbO1 and PsbO2, which are differentially expressed under varying physiological conditions. In *Arabidopsis*, PsbO2 replaces PsbO1 under stress and has been implicated in the D1 repair cycle during high-light exposure (Lundin et al., 2008; Spetea & Lundin, 2012). However, the PsbO2 isoform observed in *C. ohadii* represents a distinct structural variant characterized by its shortened β 1– β 2 loop, suggesting a different adaptation strategy.” (lines 175-180)

Lines 165-181 – the stronger PQ binding in the Qb pockets, including the potential tight interaction with the isoprenoid tail. It would be useful to mention this agreement here, and provide the reader with this work interpretation of the mechanism (hydrogen bonds (H-bonds) with His215 and Phe265, D2-Leu43) compared with the Figure 4 in Fadeeva et al (where at least Phe265 is presented, and other residues from D2). Again, can this be related to the differences in the conditions?

Answer: In agreement with Fadeeva et al. (2023), our data support the presence of strong PQ binding in the QB pocket. While Fadeeva et al. show the PQ molecule in the QB pocket in a similar position, the specific interactions are not described in the text. In their Figure 4, they show interactions (or adjacent residues) between the PQ molecule and several adjacent PsbA (D1), PsbD (D2), and PsbE (cyt b559) residues (including PHE265), but do not specify the nature of these interactions. In contrast, we performed a detailed analysis of interactions of bound PQ in the QB pocket and identified hydrogen bonds between His215 and Phe265 with the PQ headgroup, as well as hydrophobic stabilization of the isoprenoid tail involving D2-Leu43. This provides a more detailed view of PQ binding that is consistent with previous studies on the role of PQ in electron and proton transport. Only slight differences in the position of the PQ in our structure and in the structure by Fadeeva et al. could be caused by slightly different growing conditions, sample preparation and solubilization methods. Slight rotations of the amino acid side chains within the isoprenoid tail of PQ could be the result of differences in the densities and map resolution.

Based on the reviewer’s comments, we modified the text of the manuscript as follows (lines 191-209, changes in bold):

“In the recent study of *C. ohadii* (Fadeeva et al. 2023), the position of PQ in the QB pocket is comparable and residues from D1, D2, and cyt b559 forming the binding pocket were proposed. However, that study did not provide a detailed description of the nature of the molecular interactions stabilizing PQ. Our analysis revealed that these interactions involve hydrogen bonds (H-

bonds) **between His215 and Phe265 and the PQ head group. This structural finding is supported by a recent study that demonstrated that Phe265 is critical for PQ binding and efficient electron transfer in PSII (Brown et al., 2024).** ... **“Moreover, our structural findings are in agreement with the mechanistic model previously proposed by Saito et al. (2013), which highlights the importance of precise PQ positioning and its stabilization for efficient electron and proton transfer at the QB site.”**

Were the interactions in Ext. Fig. 5 here predicted from NL, HL, SHL or reproduced in all? In addition, cultures in this study were grown under C rich conditions (TAP and 2% CO₂, can this be a major driver of the differences here (see also the authors comments on lines 425-428)? In this context, it should be highlighted here that C supply from ambient air is likely to be more representative to desert crusts than 2% CO₂, so this aspect should at least be discussed, if not examined using air-levels CO₂ cultures.

Answer: The interactions in Extended Data Fig. 5 were predicted based on the NL adapted PSII structure. However, the interactions of plastoquinone with D1-residues (His215 and Phe265) are identical in Fadeeva et al. structure (PDB 8BD3), where algae cultures were grown in TAP and air-level CO₂. However, these interactions are different when compared to *C. reinhardtii* as shown in Extended Data Fig. 5. Hence, a relatively stable binding of PQ at QB site is independent of the C supply in the growth cultures but is primarily due to different architecture of QB site which appears to be a species-specific feature.

PsbR(Psb10)/PsbY – this is yet another example where the position or appearance of subunits is reported in both structures of *Ohadii*, with some differences in the interactions, and therefore can support different proposed roles. As above, readers of this work would benefit from more context (i.e. growth conditions, experimental setup) on the differences observed and whether they can resolve the role suggested here compared with the recently published structure (e.g. stabilising stacked PSII, additional shielding of b559).

Answer: We admit that this issue was not sufficiently discussed in the original version of the manuscript, which was also pointed out by the other reviewers. In the revised manuscript, the structural role of PsbR and PsbY is discussed in the broader context of the earlier structure of the PSII-LHCII supercomplex from *C. ohadii* by Fadeeva et al. (2023) and the recent structure of PSII-LHCII megacomplexes from spinach, where both subunits were also revealed (Shan et al. 2024). In addition, we have included in the revised manuscript a new Extended Data Fig 6 showing a comparison of (i) structural models of PsbR and PsbY subunits in *C. ohadii* and spinach, and (ii) the amino acid sequences with highlighted amino acids involved in specific interactions with neighboring subunits. In the revised version of the manuscript, we believe that these structural aspects have been sufficiently addressed and offer a broader view of the structural role of these two subunits.

Lines 425-428 – there is a major confusion here by the authors. That *C. reinhardtii* antenna response is also modulated by C supply, does not mean light is not playing a (major!) role. Both factors are simply the donor and acceptor side of the same electric line called linear electron flow of photosynthesis. Even in cases where some reports claim C levels can be a factor of its own (Ruiz-Sola et al. 2023), there is clearly a role for light levels under any given C level (see e.g. Figures 1, 2, 3 and many more). Similarly, *C. ohadii* antennae response cannot be claimed to be primarily driven by light when only this factor was tested. To test the effects of C levels on *C. ohadii* antennae response, the authors should have examine different C levels on the same illumination level, or at the very least compare the antennae structure here to Fadeeva et al, where air-level CO₂ was provided to the cultures (assuming acetate is no longer available in any of the cases). There are many aspects in which *C. ohadii* differs from *C. reinhardtii*, but photosynthesis in both organisms will necessarily respond to both ends of this chain. Responding to one of them, does not eliminate the response to the other.

Answer: We thank the reviewer for pointing out this ambiguity in the text. In the revised version of the manuscript (lines 505-513), we clarified that in *C. ohadii*, the light-induced modulation of LHCII antenna size appears to be independent of CO₂ availability. This is in contrast to the response observed

in the model green alga *C. reinhardtii*, where high carbon availability has been shown to lead to a reduction in LHCII antenna size under high light conditions.

Polyamines – while the observation regarding the significant increase in polyamine levels is indeed novel and interesting, I find it too preliminary to be able to extract such a detailed model of their action on PMF, which is highly speculative. To better support that, similar measurements should also be done on *C. reinhardtii* under different illuminations, in addition to testing ΔpH and other factors in *C. ohadii* upon inhibition of polyamine synthesis and/or supplementation thereof.

Answer: We fully agree that, while our findings demonstrate a significant increase in polyamine levels and suggest a potential link to photoprotective mechanisms, the current data are indeed preliminary for establishing a mechanistic model involving PMF modulation. We have therefore omitted this speculative model in the revised manuscript. Testing this mechanistic model requires a separate study, which is beyond the scope of this manuscript.

On the basis of the recommendation of Reviewer 4, in the revised manuscript we added two figures into the manuscript (Extended Data Fig. 13a and b) based on the data presented in Extended Data Table 7. The new figures clearly shows that the total polyamine content increased more than 10 times in HL when compared to NL thylakoid membranes and also which individual polyamines are related to this phenomenon.

In the revised manuscript (lines 521-530) we interpret our polyamine data within the framework proposed by Kotzabasis and coworkers, who associate elevated levels of the diamine, putrescine with the adaptation of photosynthetic organisms to high light conditions (Novakoudis and Kotzabasis, 2022).

This concept includes:

(i) a direct action of polyamines on the conformation of LHCIIs (Navakoudis et al., 2007), (ii) a direct binding of polyamines to the PSII proteins regulating the activity of PSII reaction centers (Hamdani et al., 2011), and (iii) alterations in electrical ($\Delta\psi$) and chemical (ΔpH) components of proton motive force (Ioannidis et al., 2012; Ioannidis and Kotzabasis, 2014), which regulate photochemical and non-photochemical quenching.

Manuscript No: **NCOMMS-25-07603A**

Original title: **Cryo-EM structure of photosystem II supercomplex from *Chlorella ohadii*, a green microalga with extreme phototolerance**

Reply to the reviewers' comments

Reviewer 1:

Dear Authors,

Thank you for providing detailed, point-by-point responses to all my questions and comments, and introducing changes and corrections to results of the spectroscopic analysis, all according to my suggestions. I am fully satisfied with those. I have no additional comments or suggestions.

Sincerely,

Dariusz M. Niedzwiedzki

Answer: We would like to thank the reviewer for the positive evaluation of our revised manuscript.

Reviewer 2:

My comments were addressed and I have no further comments.

Answer: We would like to thank the reviewer for the positive evaluation of our revised manuscript.

Reviewer 3:

The manuscript has been improved after revision. The authors have addressed most of my previous questions/comments in a constructive or reasonable way. Nevertheless, there are two minor points remaining to be addressed further.

1) The following is a list of chlorophyll-ligand pairs in the peripheral antennae (L-LHCII and M-LHCII) with ;coordination bond lengths having fairly large errors.

Chains 11&14, 12&15 and 13&16: CLA602 and Glu 79, CLA603 and His82, CHL609 and Glu155, CLA610 and Glu197, CLA611 and LHG615, CLA612 and Asn200, CLA613 and Gln214, CLA614 and His229.

Chains 1&4: CHL609 and Glu155, CLA610 and Glu197.

Chains 2&5: CLA610 and Glu197.

It is likely that the .eff file the authors used for structure refinement does not include the coordination bond length parameters for the above pairs as restraints during the refinement process. The errors should be fixed as much as possible by adding and applying the geometry restraints during the refinement process. An example for the coordination bond length restraint to be added in the .eff file is provided below for the author's reference.

Besides, some chlorophyll molecules have Mg atoms protruding away from (instead of toward) the ligand, such as CLA615 of chains y&Y. The wrong configuration should also be fixed too.

Answer: We would like to thank the reviewer for pointing out these small errors.

All coordination bond lengths and configurations of Mg atoms in chlorophyll molecules listed by the reviewer in his report were updated after a re-refinement with coordination bond distance set to 2.2Å.

The update includes chains of Lhcb trimers 1-2-3, 4-5-6, 11-12-13 and 14-15-16 as well as the CLA615 residues in chains Y/y. These modifications are reflected in the new model coordination file, updated to V6, in the PDB deposition site.

As requested in detail: in chains 11&14 the bond distances were updated between following pairs: CLA602 and Glu79, CLA603 and His82, CHL609 and Glu155, CLA610 and Glu197, CLA611 and LHG615, CLA612 and Asn200, CLA613 and Gln214, CLA614 and His229. In chains 12&15 those were also updated for CLA602 and Glu77, CLA603 and His80, CLA609 and Glu193, CLA611 and Asn196, CLA612 and Gln210, CLA613 and His225 and in chains 13&16 those were: CLA602 and Glu77, CLA603 and His80, CHL609 and Glu155, CLA610 and Glu193, CLA611 and LHG616, CLA612 and Asn196, CLA613 and Gln210, CLA614 and His225. In chains 1&4, the coordination of CHL609 and CLA610 was updated; in chains 2&5 the coordination of CLA610 was updated.

2) In the newly added ED Fig. 6, the amino acid sequences of *Chlorella ohadii* PsbR and PsbY do not appear to align well with those of spinach PsbR and PsbY at first sight. Do the period symbols (.) stand for the amino acid residues identical to those of *C. ohadii* PsbR? If that is the case, please explain the meanings of the period symbols and dash symbols (-) in the legends of ED Fig. 6 and the other ED figures with sequence alignment data (ED Figs. 4b, 5c, 8, 10) to avoid confusion.

Answer: We would like to thank the reviewer for pointing out this ambiguity. We have added the following text at the end of the legends of the relevant figures:

“Amino acids identical to those in the reference sequence are indicated by dots. A dash denotes a gap introduced to optimize the alignment, corresponding to the absence of an amino acid at that position.”

Reviewer 4

In the new version of the manuscript, the authors have improved the presentation of polyamine levels, which were somewhat confusing in the previous version. The interpretation of the polyamine data is now more careful and less speculative. I agree that this approach is appropriate for the manuscript's message, and that a more detailed study of the role of polyamines would constitute a separate piece of work. The authors have addressed most of my comments and provided convincing arguments for the data that were not included.

Answer: We would like to thank the reviewer for the positive evaluation of our revised manuscript.

Reviewer 5

I find the revised manuscript to be largely improved. First, the added value and comparative aspects with Fadeeva et al. are now more clearly presented, with the new insights well explained to the readers. In addition, the comparison to spinach adds an important dimension to the discussion of what may contribute to *C. ohadii* extreme resistance. Finally, some (over)statements have now been qualified to be more precise and careful. I have no additional major points to raise on this work.

Answer: We would like to thank the reviewer for the positive evaluation of our revised manuscript.